# Structural basis for the complex DNA binding behavior of the plant stem cell regulator WUSCHEL

Jeremy Sloan[1,3], Jana P. Hakenjos[2,4], Michael Gebert [2], Olga Ermakova[2], Andrea Gumiero[1,5], Gunter Stier[1], Klemens Wild[1], Irmgard Sinning [1✉] & Jan U. Lohmann [2✉]

Stem cells are one of the foundational evolutionary novelties that allowed the independent emergence of multicellularity in the plant and animal lineages. In plants, the homeodomain (HD) transcription factor WUSCHEL (WUS) is essential for the maintenance of stem cells in the shoot apical meristem. WUS has been reported to bind to diverse DNA motifs and to act as transcriptional activator and repressor. However, the mechanisms underlying this remarkable behavior have remained unclear. Here, we quantitatively delineate WUS binding to three divergent DNA motifs and resolve the relevant structural underpinnings. We show that WUS exhibits a strong binding preference for TGAA repeat sequences, while retaining the ability to weakly bind to TAAT elements. This behavior is attributable to the formation of dimers through interactions of specific residues in the HD that stabilize WUS DNA interaction. Our results provide a mechanistic basis for dissecting WUS dependent regulatory networks in plant stem cell control.

[1] Biochemistry Center, Heidelberg University, Im Neuenheimer Feld 328, 69120 Heidelberg, Germany. [2] Department of Stem Cell Biology, Centre for Organismal Studies, Heidelberg University, Im Neuenheimer Feld 230, 69120 Heidelberg, Germany. [3] Present address: BASF SE, Carl-Bosch-Strasse 38, 67056 Ludwigshafen am Rhein, Germany. [4] Present address: Celonic AG, Eulerstrasse 55, 4051 Basel, Switzerland. [5] Present address: Instituto Poligrafico e Zecca dello Stato S.p.A., Via Salaria, I-712-00138 Roma, Italy. ✉email: irmi.sinning@bzh.uni-heidelberg.de; jan.lohmann@cos.uni-heidelberg.de

Plant stem cells are embedded into specialized tissues that promote their life-long maintenance, which are called meristems. These meristems are located at the growth points of all plants, namely the shoot and root tips, as well as the vascular cylinder, to support apical-basal and lateral growth, respectively[1]. Similar to animal stem cell systems, signals controlling stem cell identity and activity within the meristem emanate from niche cells located adjacently to stem cells[2,3]. However, the cellular and molecular mechanisms of this communication are highly divergent between the two kingdoms. While cell–cell contact and secreted ligands play a central role in animals, direct cytoplasmic connections between neighboring cells, called plasmodesmata, take center stage for the maintenance of plant stem cells[4]. Interestingly, the related homeodomain (HD) transcription factors (TFs) that define the niche cells in shoot and root, namely WUSCHEL (WUS) and WUSCHEL HOMEOBOX 5 (WOX5), move to stem cells and execute their function primarily in these cells[4–6]. Consequently, there is no need for downstream niche to stem cell signaling cascades, since the DNA binding specificities of these TFs will directly dictate the repertoire of genes expressed in stem cells.

Prokaryotes and eukaryotes use different strategies to target TFs to distinct genomic locations. In bacteria, it seems to be sufficient that TFs recognize an extended DNA sequence, whereas TFs in eukaryotes typically bind to shorter DNA recognition motifs and therefore require clustering of sites to achieve specificity[7]. Numerous gene regulatory proteins of higher eukaryotes, such as leucin zipper and zinc finger TFs, bind as symmetric dimers to DNA in a sequence-specific manner, which allows each monomer to bind in a similar fashion and greatly increases the DNA binding affinity[8,9]. Consistently, the DNA recognition sequences that are bound by these TFs often are arranged as inverted or everted repeat elements. Many TFs, however, can also associate with nonidentical proteins to form heterodimers composed of two different subunits. As heterodimers are typically composed of two distinct proteins with different DNA-binding specificities, the combination of multiple TFs immensely expands the repertoire of recognized DNA sequences and greatly improves the binding specificity[10].

The eukaryotic superfamily of homeobox TFs is characterized by the presence of a HD, a short stretch of amino acids (60–66 residues) that forms a helix–loop–helix-turn-helix DNA-binding domain consisting of three alpha helices[11]. HD-TFs play a wide variety of roles in developmental and growth processes such as embryonic patterning, stem cell maintenance, and organ formation in all kingdoms of life[11–13]. In animals, the HOX TFs are the best studied family of HD proteins and specify segment identity during embryo development along the head–tail axis[14,15].

Several studies have addressed WUS DNA binding in some detail and at least three divergent sequence motifs bound by WUS, specifically sequences with a TAAT core, a G-Box like and a TGAA repeat element, have been identified[6,16–21]. The TAAT sequence was originally identified since it represents the canonical binding element for HD proteins and was subsequently experimentally confirmed to be bound by WUS by electrophoretic mobility shift assays (EMSAs) and reporter genes for multiple independent targets[6,16,17,20]. More recently, dimerization of WUS on TAAT repeats was suggested to control expression of the stem cell specific signaling factor *CLAVATA3* (*CLV3*) based on EMSA and reporter gene assays[21]. The G-Box like (TCACGTGA) motif was found in a combination of systematic evolution of ligands by exponential enrichment (SELEX) and in vivo WUS chromatin binding data derived from chromatin immunoprecipitation followed by detection by microarrays (ChIP-chip)[18]. SELEX using a recombinant WUS-HD fragment resulted in the enrichment of TCA containing sequences and the G-Box like element, which

represented an inverted repeat of TCA bases, was found to be the most overrepresented DNA sequence in chromatin regions bound by WUS[18]. Binding of WUS to this motif was confirmed by EMSA and reporter gene analysis. Lastly, the TGAA repeat motif was identified in a large-scale approach using recombinantly expressed TFs and genomic DNA in a highly parallel protein–DNA interaction screen, but has not been verified independently so far[19]. In addition to binding to multiple DNA motifs, WUS also exhibits further functional complexity by acting as transcriptional activator and repressor[17,18,21,22] and the mechanistic basis for both unusual behaviors has remained largely elusive so far.

Here, we have combined molecular, biochemical, and structural approaches to address how the WUS-HD recognizes specific DNA target sites. We find that the DNA-binding preferences of WUS-HD depend on appropriately arranged sequence motifs in a direct tandem repeat and that homodimerization is one of the key determinants to achieve high sequence specificity. We also show that disrupting the dimer interface, either on the protein level or the DNA level, severely reduces DNA-binding affinity.

## Results

### WUS has a canonical HD fold with unique structural features.
As an entry point to elucidate the mechanisms by which the WUS-HD carries out its functions, we recombinantly produced and purified a fragment containing residues 34–103 of WUS (Fig. 1a, Supplementary Fig. 1) and determined its crystal structure to a resolution of 1.4 Å (Fig. 1b, Table 1). The overall WUS-HD fold reflected that of a canonical HD structure, consisting of a three-helix bundle with an N-terminal arm. Interestingly, superposition with the structure of Engrailed (En) (PDB code 3HDD[23]) and comparative sequence analyses identified unique structural features in WUS-HD. First, the loop regions connecting the three α-helices are expanded (Fig. 1b, c). Loop region I is slightly longer compared to other HDs and is characterized by a distortion at the end of helix α1. This so called π-helix or π-bulge is typically characterized by a single amino acid insertion into an existing α-helix (Y54 in WUS) and usually correlates with a particular functional role[24,25]. Loop region II has an even longer insertion, which also extends helix α2 by an additional turn compared to En. Secondly, the N-terminal arm is anchored by docking of a tryptophan residue (W39) into a groove formed by helices α2 and α3, while a F, Y, or I residue typically performs this function in canonical HDs (Fig. 1c, d). Furthermore, this docking residue is shifted by one residue relative to the conserved DNA-contacting arginine (R38) in contrast to three residues in canonical HDs (Fig. 1c). Electrostatic surface calculations showed a large positively charged surface formed by the C-terminal recognition helix (α3) and the N-terminal arm (Fig. 1e), which represent the most conserved part of the HD (Fig. 1f). Finally, comparison with the sequence and structure of En suggested that the readout of DNA bases is mediated by the conserved residues R38, N90, and R94 of WUS-HD (Fig. 1b, c, f).

### WUS prefers tandemly arranged DNA recognition sequences.
WUS has been reported to bind to at least three divergent DNA sequences and these motifs have been proposed to be important to determine the transcriptional output of WUS[16,18,19]. Despite the obvious importance of this issue for resolving the mechanisms of WUS activity in vivo, quantitative data comparing the binding affinities to these motifs were still lacking. In order to elucidate the DNA binding preferences of WUS-HD, we therefore analyzed its interaction with three of the best studied sequences (Fig. 2a, b), namely a TAAT element from the *AG* enhancer[16], a G-Box from the *CLV1* promoter[18] and a TGAA repeat element identified in a

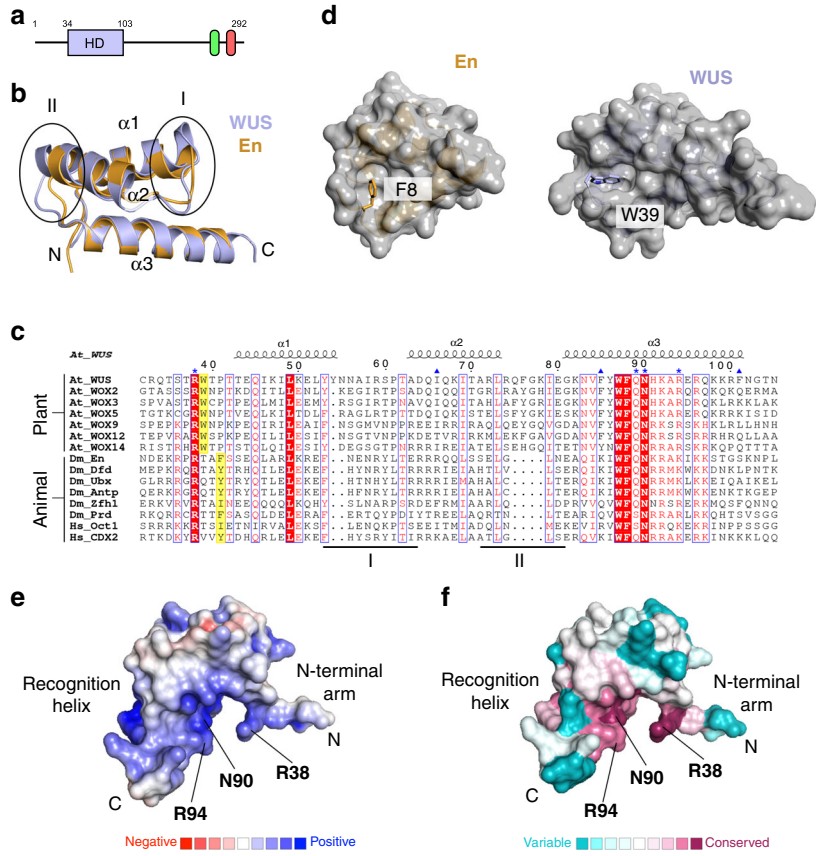

**Fig. 1 Structure and conservation of the *A. thaliana* WUS-HD. a** Domain organization of WUS from *A. thaliana*. WUS contains an N-terminal homeodomain (HD; light blue) and two short linear motifs at its C-terminus, namely the WUS-box (green) and the EAR motif (red), respectively. The domain boundaries are given in residue numbers. **b** Superimposition of WUS-HD structure (light blue) with HD from En (orange, PDB 3HDD[23]), illustrating the characteristic three-helix bundle fold of HDs. Encircled are loop regions I and II of WUS, which highlight structural differences compared to classical HDs. **c** Multiple sequence alignment of representative HDs from plants and animals. The sequences of *Arabidopsis thaliana* (*At*), *Drosophila melanogaster* (*Dm*) and *Homo sapiens* (*Hs*) were aligned using Clustal Omega and visualized with ESPRIPT. Numbering and secondary structure assignment is according to *A. thaliana* WUS. Loop regions I and II are depicted below the sequences by black lines and the anchoring residue of the N-terminal arm is illustrated by yellow boxes. Highly conserved residues are highlighted (red boxes) and residues making direct DNA-base contact (blue asterisk) and residues involved in the dimer interface (blue up-pointing triangle) are indicated. **d** Comparison of different anchoring mechanisms of the N-terminal arm for En (orange, PDB 3HDD[23]) and WUS (light blue). Surface representation of the HDs (gray) highlighting the hydrophobic pocket formed by helices α1 and α2. Amino acids responsible for fixing the N-terminal arm are shown as sticks. **e** Electrostatic surface potential (red: negative, blue: positive, contoured at ±5$k_B$T) of WUS-HD. **f** ConSurf analysis showing the degree of amino acid conservation (magenta: conserved, cyan: variable) mapped on to the surface of WUS-HD. Highly conserved amino acids typically involved in DNA base interactions are indicated.

large in vitro screen[19]. Both TGAA and G-Box harbor two atypical HD recognition motifs[26,27] that are arranged as a direct repeat and inverted repeat, respectively (Supplementary Fig. 2a, b), while the TAAT DNA only contains one typical HD recognition motif (Fig. 2b).

We employed microscale thermophoresis (MST) with a N-terminal YFP fusion of WUS-HD and 16-bp double stranded DNA probes corresponding to naturally occurring regulatory sequences containing either the TAAT, G-Box, or TGAA repeat motif. In line with earlier results[18] and in agreement with the low affinity generally reported for other HDs[11], the WUS-HD bound the TAAT probe with lower affinity compared to the G-Box containing probe with dissociation constants ($K_d$) of $10.60 \pm 1.67$ μM and $3.78 \pm 0.42$ μM, respectively; ± indicates standard deviation, $n = 3$ (Fig. 2a, b).

Intriguingly, the TGAA repeat probe was bound by WUS-HD with much higher affinity ($K_d = 0.27 \pm 0.03$ μM; ± indicates standard deviation, $n = 3$) than the other two sequences (Fig. 2a, b). As the 4-bp recognition motifs of the TGAA sequence are not significantly different from the G-Box sequence (Supplementary Fig. 2b), we

hypothesized that the relative position of recognition motifs may be a major determinant of binding specificity. To rule out any contribution of the YFP-tag to DNA-binding specificity a control measurement with YFP alone was performed, which showed no binding to DNA (Fig. 2a).

In order to compare the observed DNA-binding of WUS-HD to that of the full length (FL) protein, we expressed a fusion of WUS-FL to maltose-binding protein (MBP), and performed EMSA experiments (Supplementary Fig. 3). In accordance with our MST results, WUS-FL also exhibited divergent DNA-binding behavior, comparable to WUS-HD, when probed with TAAT, G-Box, and TGAA fluorescently labeled oligonucleotides. Consistent with the observations from the MST analysis, the TGAA repeat sequence had the highest binding affinity ($K_d = 0.36 \pm 0.06$ μM; ± indicates standard deviation, $n = 3$) of the three probes (Supplementary Fig. 3a). In addition, WUS-FL bound the G-Box probe with higher affinity compared to the TAAT containing probe ($K_d = 1.68 \pm 0.30$ μM and $K_d = 3.15 \pm 0.26$ μM, respectively; ± indicates standard deviation, $n = 3$) (Supplementary Fig. 3b, c).

**Table 1 Data collection and refinement statistics.**

| | WUS-HD | WUS-HD + G-Box DNA | WUS-HD + TGAA DNA | WUS-HD + TAAT DNA |
|---|---|---|---|---|
| *Data collection* | | | | |
| Space group | P 4₁2₁2 | C 121 | P 12₁1 | P 12₁1 |
| *Cell dimensions* | | | | |
| $a$, $b$, $c$ (Å) | 43.12, 43.12, 82.01 | 123.43, 83.07, 85.65 | 55.74, 54.34, 75.56 | 82.66, 45.91, 87.54 |
| $\alpha$, $\beta$, $\gamma$ (°) | 90, 90, 90 | 90, 112.95, 90 | 90, 107.54, 90 | 90, 102.88, 90 |
| Resolution (Å) | 38.17–1.37 (1.42–1.37) | 40.67–2.69 (2.79–2.69) | 43.38–1.58 (1.63–1.58) | 40.43–2.63 (2.72–2.63) |
| $R_{merge}$ | 0.038 (0.122) | 0.103 (1.954) | 0.040 (0.827) | 0.073 (1.878) |
| $I/\sigma I$ | 61.1 (19.9) | 11.0 (0.9) | 12.1 (1.5) | 11.6 (1.0) |
| Completeness (%) | 99.9 (99.0) | 98.1 (92.4) | 98.8 (99.1) | 99.4 (98.9) |
| Redundancy | 24.5 (16.5) | 4.8 (4.7) | 3.4 (2.6) | 4.5 (4.6) |
| *Refinement* | | | | |
| Resolution (Å) | 38.17–1.37 | 40.67–2.69 | 43.38–1.58 | 40.43–2.63 |
| No. of reflections | 16,816 (1628) | 21,855 (2051) | 59,049 (5890) | 19,403 (1907) |
| $R_{work}/R_{free}$ (%) | 18.3/21.1 | 19.5/25.1 | 18.7/22.7 | 23.7/26.3 |
| No. of atoms | 655 | 3815 | 3810 | 4053 |
| Protein | 517 | 3815 | 3501 | 4053 |
| Ligand/ion | 14 | – | 1 | – |
| Water | 64 | – | 308 | – |
| B-factors (Å2) | 22.13 | 94.48 | 38.68 | 105.49 |
| Protein/DNA | 20.03 | 94.48 | 38.41 | 105.49 |
| Ligand/ion | 52.19 | – | 57.99 | – |
| Water | 34.52 | – | 41.66 | – |
| *R.M.S. deviations* | | | | |
| Bond lengths (Å) | 0.005 | 0.010 | 0.006 | 0.004 |
| Bond angles (°) | 0.71 | 1.11 | 0.84 | 0.74 |

Values in parentheses are for highest-resolution shell.

Overall, the $K_d$ values from MST and EMSA experiments were in good agreement and deviations were mainly within measurement errors. Earlier studies had shown that WUS has the ability to homodimerize via protein domains outside the HD, which was suggested to be critical for WUS function[18,21]. However, our results showed that DNA-binding preference of WUS is dictated by the WUS-HD alone. To test whether these results reflect WUS chromatin binding behavior in living plant cells, we analyzed WUS ChIP-seq data[28] using read counts associated with the three DNA-binding motifs as a proxy for affinity (Fig. 2c). Specifically, we analyzed the probability of TAAT, G-Box, or 2xTGAA repeat motifs to be present in chromatin regions strongly bound by WUS and therefore being covered by a large number of ChIP-seq reads. Since the motifs occur in the genome at vastly divergent numbers, we converted read counts into relative binding probabilities. To this end, we plotted the relative occurrence of an individual sequence in all WUS binding peaks against the number of Chip-seq reads in a ±25 bp window around the motif. In such an analysis, very steep curves in the left part of the coordinate system indicate motifs that occur most frequently in peak regions with low ChIP-seq coverage, whereas curves that are shifted to the right indicate an association of the motif with peaks of higher ChIP-seq reads and hence are suggestive of higher affinity (Supplementary Data 1). Our analyses showed that native WUS was indeed associated with 2xTGAA repeat sequences more often than with G-Box containing genomic regions, which was followed by TTAATGG sites. Taken together, our results demonstrated that WUS strongly prefers the TGAA repeat sequence over the G-Box motif and the canonical TAAT element, both in vitro and in vivo.

**WUS-HD uses a general binding mode for different DNA sequences**. To elucidate the structural basis for these differential interactions, we solved crystal structures of WUS-HD bound to TAAT, G-Box and TGAA repeat probes to resolutions of 2.8, 2.7

and 1.6 Å, respectively (Fig. 2d). In all crystal structures of WUS-HD/DNA complexes, the unit cell contained two DNA molecules which were occupied by at least two WUS-HDs (Supplementary Fig. 4). The structure of the HD fold of each WUS molecule was not modified by the formation of the ternary complex, with an overall root mean square deviation (rmsd) before and after DNA-binding of 0.9 Å over 62 residues, although the length of the N- or C-termini vary in a context dependent manner. Interestingly, in the case of G-Box and TAAT, one of the two protein–DNA complexes in the asymmetric unit contained an additional bound HD, whereas both TGAA structures only included two HDs per DNA (Fig. 2d, Supplementary Fig. 4). In both cases, this additional molecule inserted its C-terminal recognition helix into a major groove of the DNA on the complementary strand of one of the prevalent recognition motifs, but did not make contact to the other two protein molecules. In the G-Box structure, this extra HD (Fig. 2d teal) was stabilized by crystal contacts from the neighboring complex and thus likely represents a crystallization artifact (Supplementary Fig. 4c). Furthermore, initial low resolution (>3 Å) crystal structures of the G-Box complex only ever included two HDs per DNA molecule, similar to the structure seen in Supplementary Fig. 5a.

The binding behavior in the TAAT structure was more complex. Whilst the additional HD in the TAAT complex (Fig. 2d, e light blue) bound on the opposite side of the TAAT recognition motif, one of the other HDs (Fig. 2d, e teal) contacted an unexpected DNA sequence, with less clear DNA-base interactions and an overall higher flexibility, as indicated by elevated B-factors (Supplementary Fig. 6). In order to understand the significance of these protein–DNA contacts and to delineate the critical binding regions of the TAAT sequence we performed EMSA experiments, which clearly demonstrated a change in the DNA-binding behavior of WUS-HD to the TAAT sequence probes (Supplementary Fig. 7). DNA binding to the T4C probe was largely impaired and no distinct band shifts were visible, indicating that this DNA position is important to form a stable

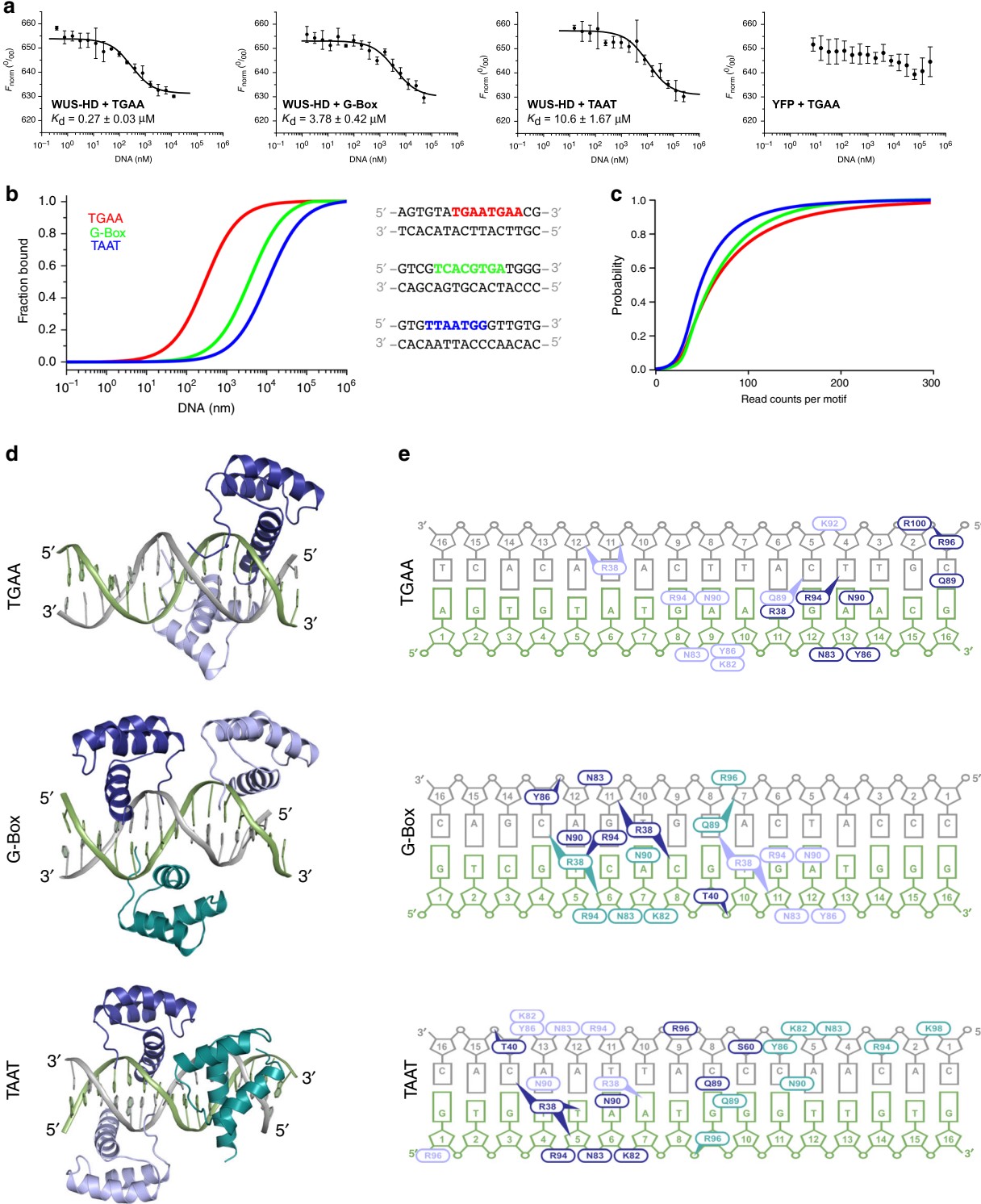

protein–DNA complex (Supplementary Fig. 7a). In contrast, the T12C probe gave a similar band shift as the wild-type (wt) TAAT sequence, suggesting no or little interference with WUS-HD binding. The band shift for the double mutant T12C, T15C was only slightly modified, suggesting that this DNA position either plays only a minor role in TAAT DNA-binding or the detection of the WUS-HD at this sequence represents an artifact of crystal packing (Supplementary Fig. 7b).

Collectively, the structural analysis and the EMSA results suggested that two HDs, which bind on opposite sides of the

TAAT recognition motif (light and dark blue in Fig. 2d, e), are crucial for an efficient interaction with the TAAT DNA. An additional HD observed in the TAAT crystal structure (teal in Fig. 2d, e) seems to be less important for DNA-binding in solution, consistent with a less defined structure and more ambiguous contact sites.

All WUS-HDs were bound to the expected 4-bp recognition motifs, except in the TAAT structure, where one molecule occupied a sequence distinct from the TAAT motif (Fig. 2e). Despite this, comparison of the ternary complex structures

**Fig. 2 Characterization of WUS-HD DNA-binding behavior in vitro, in vivo, and by crystallography. a**, DNA-binding affinity of YFP-WUS determined by MST for TGAA, G-Box, and TAAT DNA. All measurements were performed in triplicates and the respective dissociation constant ($K_d$) is indicated. A control MST reaction was performed with YFP alone in the presence of TGAA DNA. Data are means ± SEM (error bars), $n = 3$. Source data are provided as a Source Data file. **b** Comparison of the fraction bound for WUS-HD binding to different sequences of a 16-bp DNA fragment. The respective DNA recognition motifs are highlighted for TGAA (red), G-Box (green), and TAAT (blue). **c** Analysis of WUS chromatin binding in vivo by ChIP-seq. Binding probabilities of WUS to TGAATGAA (red), TCACGTGA (G-Box, green), and TTAATGG (blue) containing chromatin regions. Curves shifted to the right indicate a higher probability of a given sequence element to be associated with chromatin of high WUS occupancy and hence high affinity binding. **d** Overall structures of WUS–DNA complexes showing the mode of binding of two WUS-HD molecules per DNA for TGAA (top) and three WUS-HD molecules per DNA for G-Box (center) and TAAT (bottom). WUS-HDs are in teal, dark, and light blue and DNA-strands are in gray and green. **e** Schematic representation of DNA contact sites of WUS-HD for TGAA (top), G-Box (center), and TAAT (bottom). Ovals indicate amino acids that mainly contact a DNA base or a sugar-phosphate backbone moiety, and ovals with arrowheads specify amino acids that make multiple contacts with DNA bases and/or the sugar-phosphate backbone. Numbering of DNA bases is arbitrary starting from position 1 at each 5′-end.

revealed a very similar mode of DNA-binding for each HD; the N-terminal arm spanned the DNA minor groove, whereas the C-terminal recognition helix inserted into the major groove (Supplementary Fig. 8). Helix α3 made extensive backbone contacts, whilst both N- and C-terminal regions were engaged in establishing base-specific contacts. In almost all WUS-HD molecules, the N-terminal arm inserted R38 into the minor groove to hydrogen bond with base pairs and typically specified a thymine at the −2 position[29] (Fig. 2e, Supplementary Fig. 8). The hydrogen bond donor–acceptor pattern was neither specific to the sense- or antisense-strand, however, the readout by R38 was mediated by base pair recognition and may have also be dependent on DNA shape[30]. The majority of DNA contacts were established by major groove interactions (Fig. 2e, Supplementary Fig. 5b), involving extensive backbone contacts as well as base pair recognition.

The readout of bases in the major groove was mediated by the conserved residues Q89, N90, and R94 (Figs. 1f, 2e, and Supplementary Fig. 5b). N90 that specified adenine (position 0), crucial for HD binding, appeared as most relevant[23,31]. R94 favored a guanine at position −1 from the adenine (Supplementary Fig. 2b), in agreement with the specificity of atypical HDs[26,27]. Interestingly, in the TAAT structure position −1 was an adenine, similar to the DNA recognition motif of typical HDs; thus, in this complex R94 was not involved in base recognition and instead contacted the sugar phosphate backbone (Fig. 2e, Supplementary Fig. 8). The role of Q89 was less clearly defined by the structures; in some cases, it did not interact directly with DNA and in others it contacted a base at position +2 or +3 on either side of the double strand, consistent with the idea that the conserved Q89 promotes the recognition of bases at these positions[32,33].

Residues K82, N83, and Y86 formed a cluster which bound consecutive phosphate groups of the DNA backbone. K92, R96, and R100 were also involved in backbone contacts, although to a lesser extent as these interactions were not present in all structures and thus presumably were dependent on protein–protein interactions (Fig. 2e, Supplementary Fig. 5b). Importantly, all side-chains involved in the readout of base pairs were among the most highly conserved residues of the WUS-HD (Figs. 1c and 2e). In addition, the observed protein–DNA contacts were consistent with results obtained with other HDs, where specific interactions are established with a 4–7 bp DNA binding site[26,27]. Most other conserved amino acids without indicated functions appeared to have structural roles in maintaining the overall HD fold (Fig. 1c).

**WUS-HD prefers the atypical TGAA over the typical TAAT motif.** Having identified the DNA recognition preferences of WUS, we compared the binding mode with typical and atypical HDs from metazoans with similar interaction motifs (Fig. 3). The

"typical" Antennapedia (Antp) HD binds to a core TAAT motif (PDB code 4XID[34]) as found in our *AG* derived TAAT probe. The residues involved in establishing base-specific contacts are conserved, however there are notable differences in the interactions formed by Antp-HD and WUS-HD (Fig. 3b). Commonly, arginine (or lysine) as residue R2 or R3 enables specific read-out of the adenine in the −1 position[29,35], the hallmark of the typical recognition motif. In contrast, the equivalent residues in WUS-HD (T35 and S36) did not form this base-specific contact. However, the N-terminal arm of WUS-HD still contacted adenine −1 via R38, equivalent to the highly conserved R5 that conventionally reads out the −2 position only (Fig. 3a, b).

The binding of the "atypical" Extradenticle (Exd) HD (PDB code 2R5Y[30]) to a core TGAT motif, although very similar in sequence, also shows some differences in the hydrogen bond pattern compared to WUS-HD bound to the G-Box probe (Fig. 3b). Notably, the conserved R38 of the N-terminal arm in our structure neither contacted the −1 nor the −2 position of the sense-strand. Instead, R38 bound to positions −2 and −3 of the antisense-strand, which highlighted the broader specificity of HDs at position −1 and −2 of the core recognition motif[26,27]. However, the guanine at position −1 was specified by R94 in the C-terminal recognition helix, following the usual mechanism for HDs contacting atypical recognition motifs (Fig. 3). Taken together, our crystal structures indicated that WUS-HD is able to establish the canonical base-specific contact with guanine in position −1 but not adenine. As the nucleotide base in this position is the main determinant in preferential recognition of typical or atypical motifs, this would suggest that WUS-HD prefers binding to atypical motifs. Regardless, WUS-HD is still able to form specific interactions with the typical TAAT motif, likely reflecting the inherent broad specificity of HDs for DNA sequence recognition[26,27].

**WUS-HD binding specificity depends on DNA shape.** We noticed that position +1 of the DNA motif made no hydrogen bonds to the protein in any of our structures and therefore we analyzed whether WUS had any base-preference at this position (Fig. 4a). Intriguingly, MST experiments showed a strong preference of WUS-HD for A or T, with binding to A containing probes roughly twofold stronger than to probes with a T at this position (Supplementary Fig. 9a, b). In contrast, probes with C or G were bound less tightly, decreasing affinity by ~7-fold and ~26-fold, respectively (Supplementary Fig. 9c, d). Interestingly, the recognition motifs in the TGAA repeat sequence each have adenine at position +1, consistent with the observation that this sequence was bound with a higher affinity than the other two crystallized variants (Fig. 2a, b).

Why does the +1 DNA motif position have such a strong influence on WUS-HD affinity, despite the fact that this base is not involved in hydrogen bond interactions? Since DNA shape

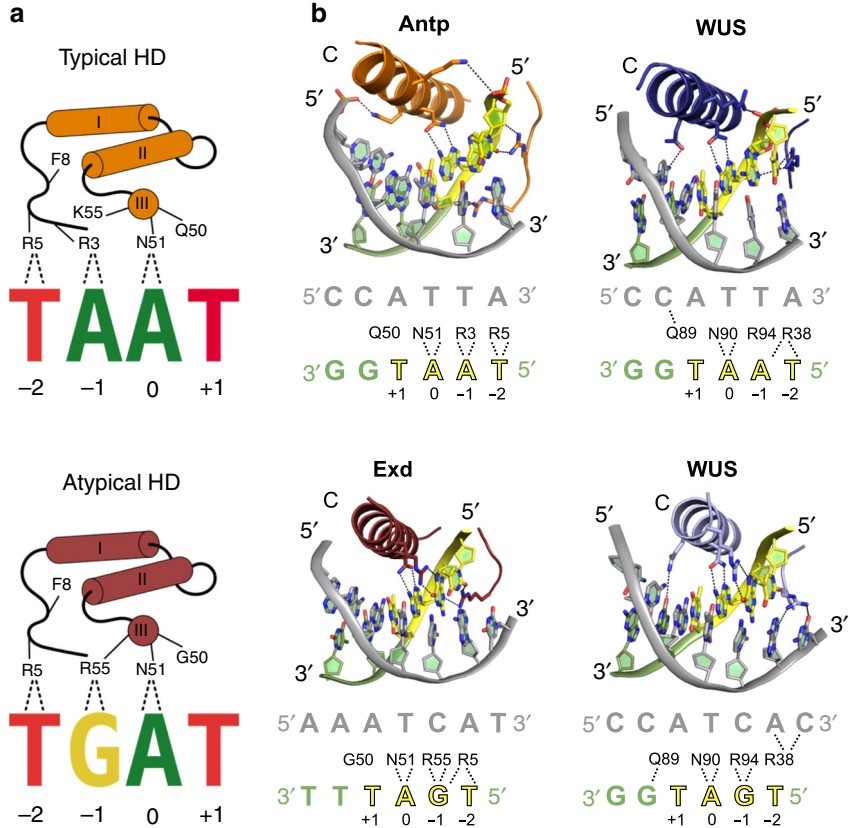

**Fig. 3 Comparison of sequence specificity with typical and atypical HDs. a** Schematic representation of DNA sequence specificity for typical and atypical HDs. The DNA base-recognition details are depicted for a typical HD in orange (top) and an atypical HD in red (bottom). Note the specific read-out of the adenine at position 0 by N51, characteristic for HD proteins. The numbering of residues is according to the Antp-HD[34] **b** DNA base-recognition details of WUS compared to other HDs. Top, showing hydrogen bond interactions of Antp (orange, PDB 4XID[34]) and WUS with the same typical core DNA recognition motif (yellow). Bottom, showing hydrogen bond interactions of Exd (red, PDB 2R5Y[30]) and WUS with the same atypical core DNA recognition motif. Diagrams below each cartoon representation summarize DNA-base contacts made by each HD.

can have a substantial effect on specificity and affinity of HD–DNA complexes[36], we computationally investigated potential structural differences in the DNA sequences experimentally tested. To this end, we used the DNAShape tool[37] to predict the intrinsic conformation of unbound DNA probes differing in the +1 position focusing on minor groove width (MGW) (Fig. 4b). Consistent with the overall similarity of the sequences, the predicted MGW profiles are similar in all cases with two MGW minima occurring around the different nucleotides at the +1 position. Interestingly, these minima spatially coincided with the position of the WUS R38 side chain insertion for contacting thymine 7 and thymine 11 (Fig. 4b, c).

Our analysis showed that the local MGW minima for the 3xTGAA and 3xTGAT sequences are much more pronounced than in the 3xTGAC and 3xTGAG sequences. The 3xTGAA sequence, which had the highest affinity of the four sequences ($K_d = 0.06 \pm 0.01\,\mu M$; $\pm$ indicates standard deviation, $n = 3$), showed two strong minima that overlapped best with the binding position of WUS R38 (Fig. 4b). In contrast, the 3xTGAT sequence, which had a slightly weaker affinity ($K_d = 0.12 \pm 0.01\,\mu M$; $\pm$ indicates standard deviation, $n = 3$), also exhibited two strong minima, however, they were shifted to position +2. In addition, the 3xTGAC and 3xTGAG sequences had even weaker affinities ($K_d = 0.39 \pm 0.05\,\mu M$ and $K_d = 1.42 \pm 0.17\,\mu M$, respectively; $\pm$ indicates standard deviation, $n = 3$), consistent with the local MWG minima being in a different position and a less narrow minor groove. Consequently, the DNA Shape tool predictions suggested that the 3xTGAC and 3xTGAG sequences are less well pre-organized for WUS DNA-binding and

require larger conformational changes compared to the 3xTGAA and 3xTGAT sequences.

Besides the widely recognized hydrogen bond interactions of specific bases, hydrophobic contacts can also be an important determinant for protein–DNA specificity[38]. Analysis of the DNA contacts in the structures of TGAA and G-Box, where the +1 motif position was an adenine or thymine, respectively, revealed that hydrophobic residues of WUS-HD made contact to the C5 methyl group of a thymine base (Fig. 4d). In the G-Box crystal structure, Y86 formed Van der Waals interactions with thymine of the +1 position of the TGAT motif. In addition, the aliphatic chain of K82 interacted with Y86 and thus contributed to the local hydrophobic environment. In contrast, in the TGAA structure A93 contacted the thymine from the complementary strand, which base paired with the adenine of the +1 position (Fig. 4d). Thus, despite the fact that the +1 DNA motif position was not involved in base-recognition via hydrogen bonds with WUS-HD, the bases at this position had a substantial influence on local DNA conformation and, together with Van der Waals contacts of hydrophobic side chains from WUS-HD, led to a strong preference for A/T over G/C. These findings were also consistent with the experimental observation that the G-Box probe, which contained TGAG and TGAT motifs, was bound with much lower affinity than the TGAA probe, which has two TGAA motifs (Fig. 2a, b). Furthermore, thymine is the most common base at position +1 in typical recognition motifs and correlates with the presence of an aliphatic residue contacting this position[26,27].

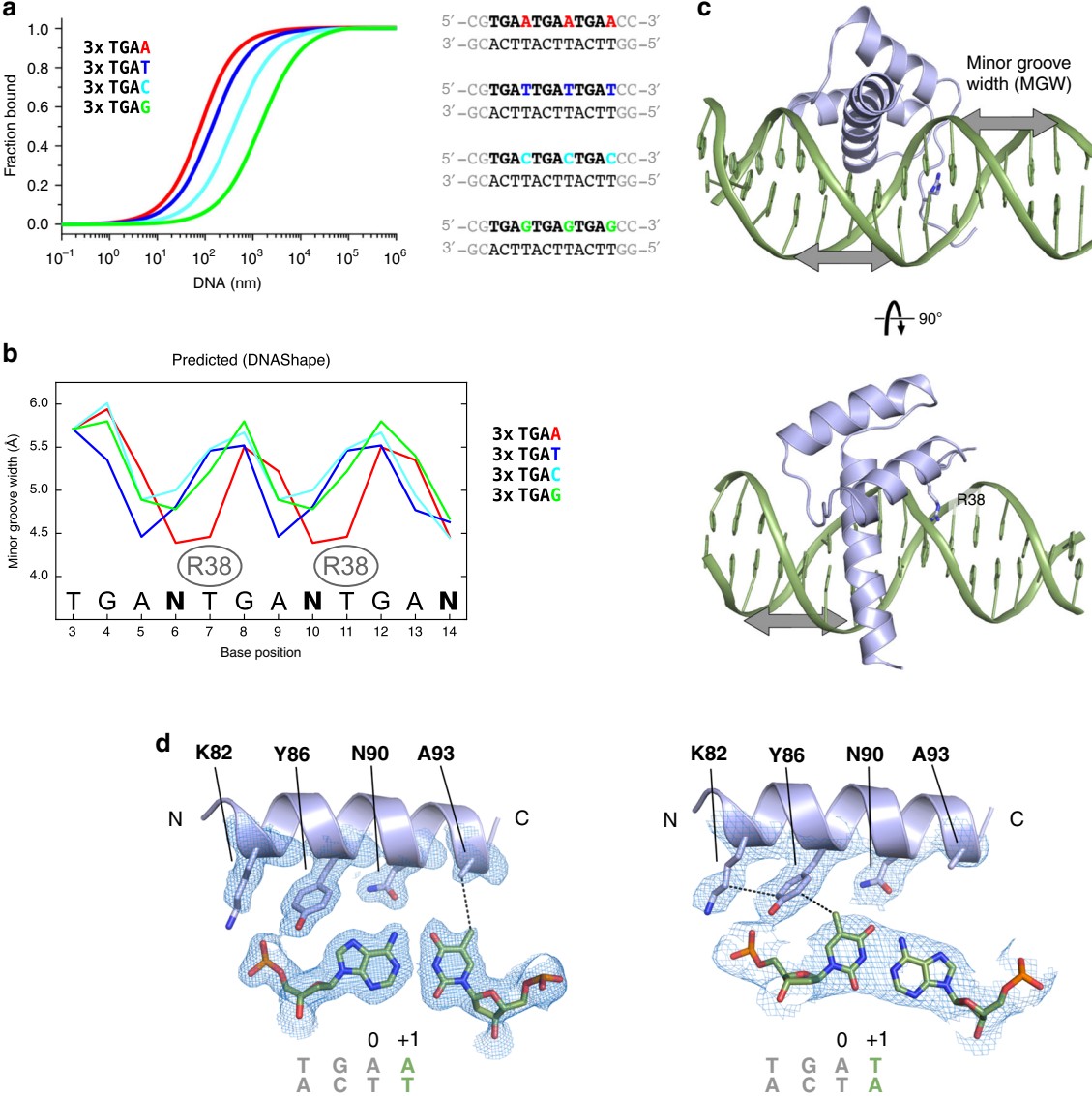

**Fig. 4 Molecular basis for preference of the +1 recognition motif position. a** MST-analysis of WUS sequence recognition specificity for DNA position +1, based on an atypical recognition motif. The tandem recognition motif is in bold letters and position +1 is adenine (red), thymine (blue), cytosine (cyan), and guanine (green). **b** Predicted minor groove width (MGW) profiles for atypical DNA sequences differing only in the +1 position of the WUS-HD recognition sequence. The color scheme is the same as in (**a**) and the binding position of WUS R38 inserting into the minor groove is indicated. The DNA sequence is shown at the bottom, where *N* represents any of the four nucleotides (A,T,C,G). **c** Structure of WUS (light blue) bound to DNA (green) highlighting the insertion of R38 into the minor groove. The minor groove width (MGW) is indicated by gray arrows. **d** Structural basis of adenine (TGAA, left) and thymine (G-Box, right) preference at the +1 position. Shown are the residues of the WUS recognition helix making hydrophobic contacts with the C5 methyl group of thymine. The conserved Asn90 residue is shown as a reference and the 2Fo–Fc electron density maps (blue mesh) are contoured at 1.0σ.

**The WUS-HD undergoes DNA-mediated dimerization.** One of the surprising findings of our crystallization experiments was that two WUS-HDs were found to bind every DNA molecule, even though the canonical TAAT motif is usually only bound by a single HD[26,27]. In addition, we observed that irrespective of the probe sequence, the two WUS-HD molecules are engaged in protein–protein contacts (Fig. 5a, Supplementary Fig. 4) even though multi angle light scattering (MALS) demonstrated that the WUS-HD was monomeric in solution (Supplementary Fig. 10a). Interestingly, these DNA-bound dimers had a unique relative orientation in all structures. Bound to the G-Box probe, the two monomers were positioned head-to-head on the same side of the DNA and had almost identical binding features, probably due to the palindromic nature of the DNA recognition sequence and the

negligible interaction between them (Figs. 2d, 5a, and Supplementary Fig. 11a). In contrast, the two HD molecules interacting with the TAAT and the TGAA repeat probes were on opposite sides of the DNA and formed specific protein-protein modifications between each other (Figs. 2d and 5a). One of the WUS-HD molecules bound to the TGAA probe made additional DNA contacts through stabilization of the helix α3 C-terminus by the other WUS molecule (Fig. 2e, Supplementary Fig. 11b), which might explain the higher affinity for the TGAA sequence (Fig. 2a, b). In particular, R96 and R100 of the recognition helix established new contacts to the DNA-backbone, not observed in any of the other WUS–DNA complexes. Similarly, extensive protein–protein interactions with the HD bound to the typical core TAAT sequence likely allowed an additional WUS-HD

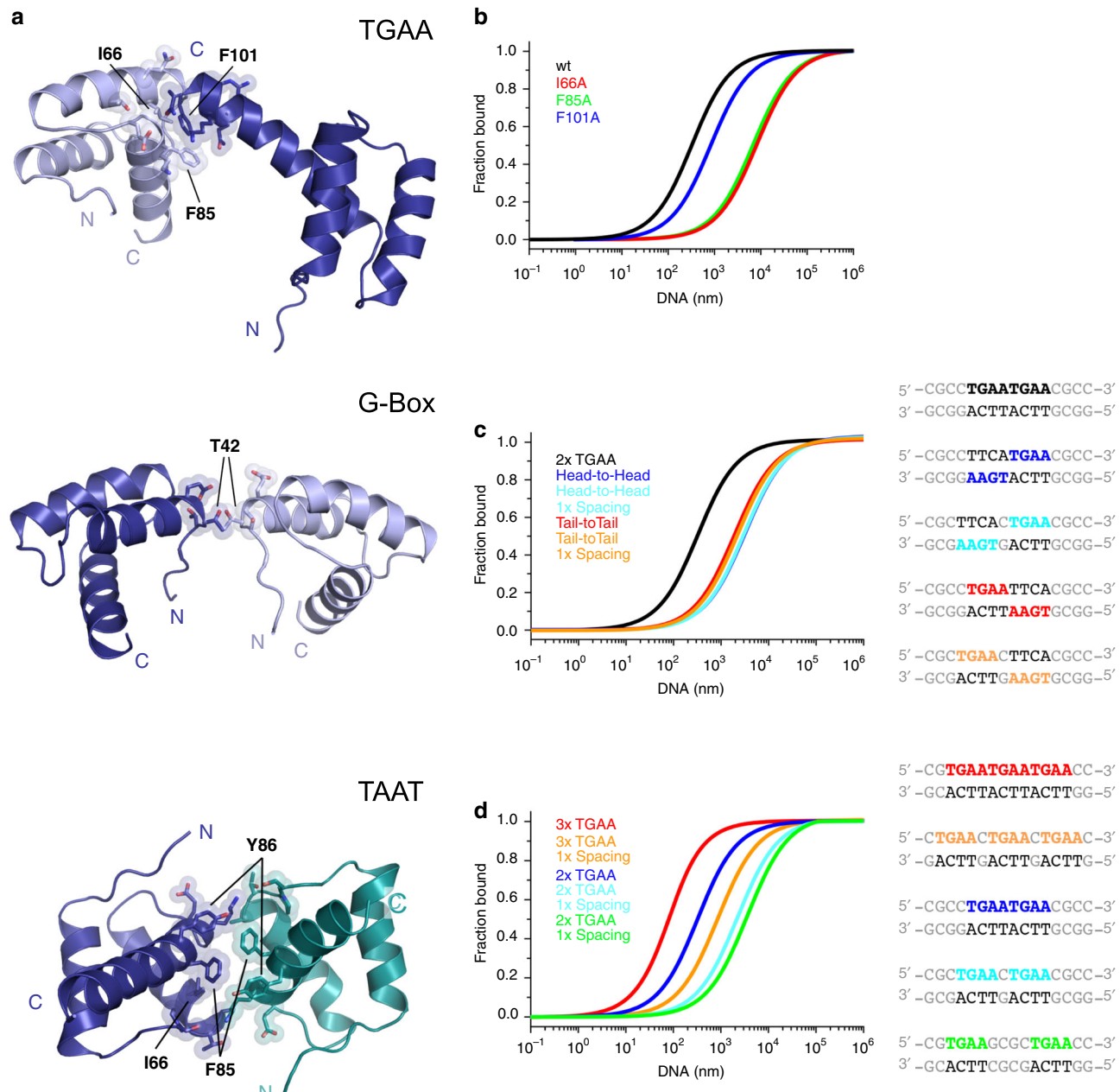

**Fig. 5 DNA sequence specificity depends on WUS-HD dimerization. a** DNA-facilitated protein interactions between individual HDs of WUS bound to TGAA (top), G-Box (center), and TAAT (bottom). For clarity DNA was omitted and side chains mediating the protein–protein interface are shown. Amino acids contributing most to the buried surface area are indicated and colors schemes are related to Fig. 2. **b** MST-analysis of the DNA-facilitated WUS dimerization interface. Single point mutants were introduced (I66A in red, F85A in green and F101A in blue) and binding was quantified for a 2xTGAA motif. **c** MST-analysis of orientation preferences for WUS binding toward different arrangements of a 2xTGAA DNA motif. The head-to-head arrangement is in blue and the tail-to-tail arrangement is in red, whilst the same arrangements with a 1 bp spacer are shown in cyan and orange, respectively. As a reference, the binding towards the tandem repeat 2xTGAA DNA motif is shown in black. **d** MST-analysis of spacing preferences for WUS DNA-binding activity. Binding affinity was measured for three TGAA recognition motifs (red) and with a 4 bp spacer (orange), and for two TGAA recognition motifs (blue), with a 1 bp spacer (cyan) and a 4 bp spacer (green).

molecule to occupy an unexpected position in the TAAT structure (Figs. 2d and 5a). However, in this case, the interaction with the DNA was less important and the structure was not well resolved in the electron density, as indicated by elevated B-factors (Supplementary Fig. 6, Supplementary Fig. 11c). In contrast, in the other, likely more relevant configuration, the two WUS-HD molecules did not exhibit any protein–protein interface, but individually formed a stable protein–DNA complex with the TAAT probe (Fig. 2d, Supplementary Fig. 7).

**Cooperative binding determines WUS-HD sequence specificity.**
Further analysis of the DNA-binding activity of WUS-HD by MST measurements clearly demonstrated a gain in binding affinity with increasing number of recognition motifs, indicating that the binding of multiple HD molecules per DNA molecule occurs in solution as well as in our crystal structures (Supplementary Fig. 12a). Although in this experimental setup, the derived $K_d$ values are not directly comparable due to the variation in the number of binding sites, the increase in affinity from one

(1xTGAA, $K_d = 10.50 \pm 2.30\,\mu M$; $\pm$ indicates standard deviation, $n = 3$) to two binding sites (2xTGAA, $K_d = 0.30 \pm 0.04\,\mu M$; $\pm$ indicates standard deviation, $n = 3$) was still higher than expected for two independent binding sites ($K_d \approx 2–4\,\mu M$). Thus, we hypothesized that this must be a cooperative effect due to favorable interactions between the protein molecules. Interestingly, the affinity of an ideal 2xTGAA DNA repeat was similar to that of the naturally occurring TGAA repeat probe from the *CLV1* locus ($K_d = 0.27 \pm 0.03\,\mu M$; $\pm$ indicates standard deviation, $n = 3$) (Fig. 2a, Supplementary Fig. 9e), suggesting the binding of two WUS-HD molecules per DNA as seen in our TGAA crystal structure (Fig. 2d). Indeed, the *CLV1* derived sequence has two TGAA and one TGTA motif (Fig. 2b), demonstrating the importance of adenine at the 0 position, crucial for HD binding[23,31].

To test the relevance of cooperativity for chromatin binding of WUS in vivo, we quantified reads of our ChIP-seq data aligning to sequences containing one, two, or three TGAA recognition motifs (Supplementary Fig. 13a). Consistent with the increase in binding affinity seen by MST, multiple TGAA repeat motifs were bound by WUS much more frequently compared to individual TGAA sequences, demonstrating that cooperative binding is a relevant mechanism for WUS chromatin interaction in vivo.

In order to assess the complex stoichiometry of WUS-HD/DNA complexes we performed MALS analysis with the same DNA probes containing tandem TGAA recognition motifs (Supplementary Fig. 10b–e). However, the determined molecular mass ($MM_{calc}$) for the complex fraction was always approximately 9 kDa lower than the expected theoretical molecular mass ($MM_{theo}$), if all recognition motifs were occupied. This suggested the absence of one WUS-HD monomer in the final protein–DNA complex and could be due to a dilution effect during gel filtration.

To unambiguously determine the number of binding events and to probe for cooperativity, we used isothermal titration calorimetry (ITC) to quantify the binding thermodynamics of WUS-HD with a 2xTGAA recognition motif (Supplementary Fig. 12b). In line with our expectations, binding of the 2xTGAA DNA could be fit best by a sequential binding model, indicating two binding events ($K_{d,1} = 1.24 \pm 0.14\,\mu M$ and $K_{d,2} = 0.82 \pm 0.07\,\mu M$; $\pm$ indicates standard deviation, $n = 3$) with positive cooperativity and a protein to DNA stoichiometry of 2:1.

**Dimerization drives cooperative binding of repeat motifs.** To mechanistically dissect the positive cooperativity for WUS-HD binding to atypical TGAA repeat sequences, we investigated the protein–protein interactions between the DNA-bound HD dimers observed in our crystal structures in more detail (Fig. 5a). Although the interaction surface area between the WUS-HD molecules was very small in all cases, covering only 2–6% (90–290 Å$^2$) of the solvent accessible surface area, a few hydrophobic residues (I66, F85, and F101) were notably more buried in both the TAAT and TGAA crystal structures (Fig. 5a). To functionally test the contribution of the dimerization interface to the DNA-binding activity of WUS, we independently substituted these residues with alanine. MST analysis revealed a reduced DNA-binding affinity of all three WUS-HD variants (Fig. 5b), supporting our hypothesis that high affinity DNA-binding to TGAA repeat probes requires WUS homodimerization. Interestingly, only positions I66 and F85, lining the interface of helices α2 and α3 of WUS-HD, are conserved within the WOX family, suggesting that F101 may represent a specific feature of WUS compared to other WOX members (Fig. 1c). Consistent with these findings, we observed that the F101A mutation only reduced binding affinity by a factor of about three ($K_d = 0.80 \pm 0.12\,\mu M$; $\pm$ indicates standard deviation, $n = 3$) compared to >20

($K_d = 8.20 \pm 0.84\,\mu M$; $\pm$ indicates standard deviation, $n = 3$) and >20 ($K_d = 6.81 \pm 0.87\,\mu M$; $\pm$ indicates standard deviation, $n = 3$) for the I66A and the F85A mutations, respectively (Fig. 5b, Supplementary Fig. 9o–q).

To test whether these substitution alleles indeed modify the dimerization status rather than indirectly reducing DNA binding affinity by more globally affecting WUS-HD structure, we analyzed the interaction between the TGAA direct repeat probe and the mutants by EMSA experiments (Supplementary Fig. 14). In accordance with our MST results, all variants exhibited divergent DNA binding behavior compared to wild-type WUS-HD when probed with the 2xTGAA repeat sequence. Consistent with the observations from the MST analysis, DNA binding of the I66A and F85A variants was largely impaired and no distinct band shifts were visible, suggesting that these conserved residues may play an important role for the overall fold of WUS-HD rather than only mediating dimerization (Supplementary Fig. 14b, c). In contrast, the F101A variant still bound DNA with reasonable affinity as observed in MST, but the resulting complex was predominantly monomeric, in comparison to the mostly dimeric form observed with wild-type WUS-HD (Fig. 5b, Supplementary Fig. 14a, d).

Therefore, these results confirmed that WUS-HD forms a homodimer upon DNA-binding, where the interaction between the monomers is scaffolded by DNA and limited to a few amino acid contacts with an important role for F101. In addition, the newly identified dimerization sites of WUS greatly contribute to the cooperative DNA-binding of tandemly arranged TGAA recognition motifs, such as the ones observed in the TGAA direct repeat or *CLV1* derived TGAA probes (Fig. 2a, b).

Since the arrangement of recognition motifs is likely to influence WUS-HD binding affinities for all probes (Fig. 2a, b), we further examined how WUS-HD DNA-binding depends on the orientation or spacing of two identical core recognition motifs using the TGAA interaction as a model. Interestingly, changing the relative position of the TGAA core recognition motif from a direct tandem repeat into an inverted (tail-to-tail) or everted (head-to-head) repeat configuration on opposite strands led to a drastic decrease in binding affinity by about ~10-fold (Fig. 5c). The affinity of the head-to-head sequence probe ($K_d = 3.17 \pm 0.35\,\mu M$; $\pm$ indicates standard deviation, $n = 3$) was similar to that of the naturally occurring G-Box probe ($K_d = 3.78 \pm 0.42\,\mu M$; $\pm$ indicates standard deviation, $n = 3$) from the *CLV1* locus, consistent with the observation that this sequence was bound with a lower affinity compared to the TGAA probe (Fig. 2a, b). This reduction was also observed when changing the orientation from a head-to-head arrangement to a tail-to-tail arrangement ($K_d = 1.90 \pm 0.20\,\mu M$; $\pm$ indicates standard deviation, $n = 3$) (Fig. 5c, Supplementary Fig. 9h, j). To test whether these observations are relevant for WUS chromatin binding in vivo, we again mined our ChIP-seq data. In accordance with the MST results, the binding probabilities showed a clear correlation with the orientation of two TGAA motifs (Supplementary Fig. 13b). The direct TGAA repeat motif was bound significantly more often compared to the head-to-head and the tail-to-tail configuration, which both had similar read distributions, consistent with the binding affinities determined by MST. These results confirmed that high affinity binding of WUS-HD to direct TGAA repeat sequences is dependent on the protein–protein interactions observed in our crystal structure, and that these interactions are highly relevant in vivo.

To test this further, we analyzed direct tandem repeat sequences with variable spacing between the TGAA motifs by MST (Fig. 5d). In line with our hypothesis that interactions between neighboring WUS-HD molecules are required for high affinity binding, we observed a substantial reduction in

DNA-binding affinity when we separated the TGAA motifs. Additional spacing by one nucleotide led to a reduction by ~7-fold ($K_d = 2.21 \pm 0.25\,\mu M$; ± indicates standard deviation, $n = 3$) and ~15-fold ($K_d = 0.84 \pm 0.12\,\mu M$; ± indicates standard deviation, $n = 3$) for 2xTGAA and 3xTGAA respectively (Fig. 5d, Supplementary Fig. 9l, m). Increasing the spacer length up to four nucleotides ($K_d = 3.58 \pm 0.44\,\mu M$; ± indicates standard deviation, $n = 3$) did not lead to a more pronounced effect. Notably, this effect was not observed when we introduced a spacer between motifs situated on different DNA strands of inverted and everted repeat probes (Fig. 5c), a motif arrangement that does not allow protein–protein interactions to begin with. Consistent with the observations from the MST analysis, the ChIP-seq data also showed a reduction in binding probability when two TGAA motifs were separated by an additional nucleotide (Supplementary Fig. 13c). Taken together with the cooperativity shown by ITC, these results strongly suggested that stabilizing protein-protein interactions between WUS-HDs promote high affinity binding to DNA containing direct repeats of tandemly arranged TGAA recognition motifs.

**Base specific contacts are crucial for sequence specificity.** Characterization of the WUS-HD DNA-binding specificity has shown that WUS-HD prefers atypical TGAA repeat sequences, while typical TAAT elements were bound less efficiently (Figs. 2b and 3b). Hence, we wanted to identify the mechanisms responsible for this behavior and test whether we could reprogram the DNA-binding preferences of WUS from an atypical TGAA motif to a typical TAAT motif. In atypical HDs (e.g., Exd[30]), an arginine (R94 in WUS) reads out the guanine at position −1 of the DNA recognition sequence. However, in typical HDs this residue is commonly a lysine, which contacts the sugar-phosphate backbone (Fig. 6a). In addition, typical HDs (e.g., Antp[34] and En[23]) usually contain one or two positively changed residues at their N-terminal arm that specify an adenine at position −1 of the DNA recognition motif. In WUS these residues are T35 and S36, which were either not visible in the crystal structures of the WUS/DNA complexes or not involved in DNA contacts.

In order to test the contribution of these residues to the DNA binding preferences of WUS-HD, we therefore separately substituted T35 and S36 to arginines to promote the recognition of the typical TAAT motif via the adenine in the −1 position (Fig. 6a). In addition, R94 was substituted to lysine to abolish the interaction with guanine at position −1 of the atypical TGAA motif. To probe the DNA sequence specificity of these modified WUS-HD variants, EMSA experiments were performed, which clearly demonstrated a change in the DNA-binding behavior of WUS-HD to TGAA and TAAT sequence probes (Fig. 6b). Both the T35R and the S36R variant gave similar band shifts as wild-type (wt) WUS-HD when probed with the TGAA repeat sequence, suggesting no interference with binding of atypical recognition motifs. However, when probed with the TAAT sequence the band shift pattern for these variants changed compared to the WUS wt and shifted to a predominantly monomeric complex, characteristic for typical HDs binding to the TAAT recognition motif. The R94 variant on the other hand, had no apparent influence on DNA-binding to the TAAT probe compared to WUS wt (Fig. 6b). In contrast, binding to the TGAA repeat probe was significantly impaired and no distinct band shifts were visible. A triple WUS-HD RRK mutant (T35R, S36R, and R94K) was then analyzed for a synergetic effect on WUS DNA-binding specificity. To our surprise, combining all three mutations resulted in an intermediate binding to the TGAA probe, suggesting that the T35R and S36R mutations had restored the binding ability that had been diminished by the individual

R94K mutation (Fig. 6b). The binding behavior to the TAAT probe was similar to the T35R and S36R mutants, demonstrating the preference for typical recognition motifs in this configuration.

To quantitatively delineate the observed changes in DNA-binding specificity, we analyzed the interaction between the TGAA and TAAT probes and the individual WUS-HD mutants by MST (Fig. 6c). Consistent with the observations from the EMSA experiments, the T35R ($K_d = 0.54 \pm 0.07\,\mu M$; ± indicates standard deviation, $n = 3$) and S36R ($K_d = 0.32 \pm 0.03\,\mu M$; ± indicates standard deviation, $n = 3$) variants had similar binding affinities to the TGAA probe as the wild-type ($K_d = 0.27 \pm 0.03\,\mu M$; ± indicates standard deviation, $n = 3$) (Supplementary Fig. 9r, t). Similarly, the ~40-fold decrease in binding affinity for the R94K ($K_d = 11.00 \pm 1.62\,\mu M$; ± indicates standard deviation, $n = 3$) variant agreed very well with the drastically impaired DNA-binding ability seen in EMSA (Fig. 6b, c). Interestingly, the same mutant enhanced binding to the TAAT probe by a factor of two ($K_d = 4.59 \pm 0.58\,\mu M$; ± indicates standard deviation, $n = 3$) compared to WUS-HD wt, while the T35R and the S36R mutations both increased binding affinity by a factor of six ($K_d = 1.67 \pm 0.20$ $\mu M$ and $K_d = 1.67 \pm 0.17\,\mu M$, respectively; ± indicates standard deviation, $n = 3$) (Fig. 6c, Supplementary Fig. 9u–w). Collectively, the results from the EMSA and MST experiments suggested that the high specificity of WUS-HD for the atypical TGAA repeat sequence was strongly dependent on R94, as substituting this residue to lysine significantly compromised DNA-binding affinity. In contrast, the introduction of arginines at the N-terminal arm (T35R and S36R) improved the recognition of TAAT sequences by WUS-HD.

## Discussion

The HD transcription factor WUS plays key roles for plant development with at least four distinct and essential subfunctions: Firstly, regulation of stem cell identity in the shoot apical meristem (SAM), where it is required for non cell-autonomous induction and maintenance of stem cell fate via stimulation of cytokinin and suppression of auxin signaling[2,4,6,17,28,39]. Secondly, regulation of floral patterning through direct regulation of differentiation genes[16]. Thirdly, coordination of cell identity specification during female gametophyte development[40]. And lastly, cell differentiation during anther development[41]. Consistent with a broad functional portfolio, WUS has been shown to bind to a diverse set of DNA motifs in the regulatory regions of downstream genes involved in SAM development and signaling processes[16–21,28]. However, current in vitro DNA binding data have little predictive value when it comes to understanding WUS activity in vivo, since a clear correlation between the occurrence of specific DNA motifs with functional classes of response genes or activation or repression of targets has not emerged so far.

In the present study, we quantified the binding affinity of WUS-HD toward different known DNA sequences and performed structural and biochemical analyses to gain insight into the mechanism underlying its binding diversity. The crystal structure of WUS revealed a canonical HD fold with unique structural features compared to classical HD proteins found in metazoans, including elongated loop regions and a divergent mechanism for positioning the N-terminal arm. While most of these differences also apply on a sequence level to other WOX proteins, WUS is the only member where the specific insertion of a tyrosine residue at position 54 leads to a bulged-out helix or π-helix, typically correlated with functional sites in a protein[24,25]. WUS uses a general mode of DNA-binding to all sequence motifs and in all cases the DNA is bound as dimer, even the classical TAAT motif usually only bound as monomer by all HD TFs analyzed so far. Interestingly, the DNA bound dimers have

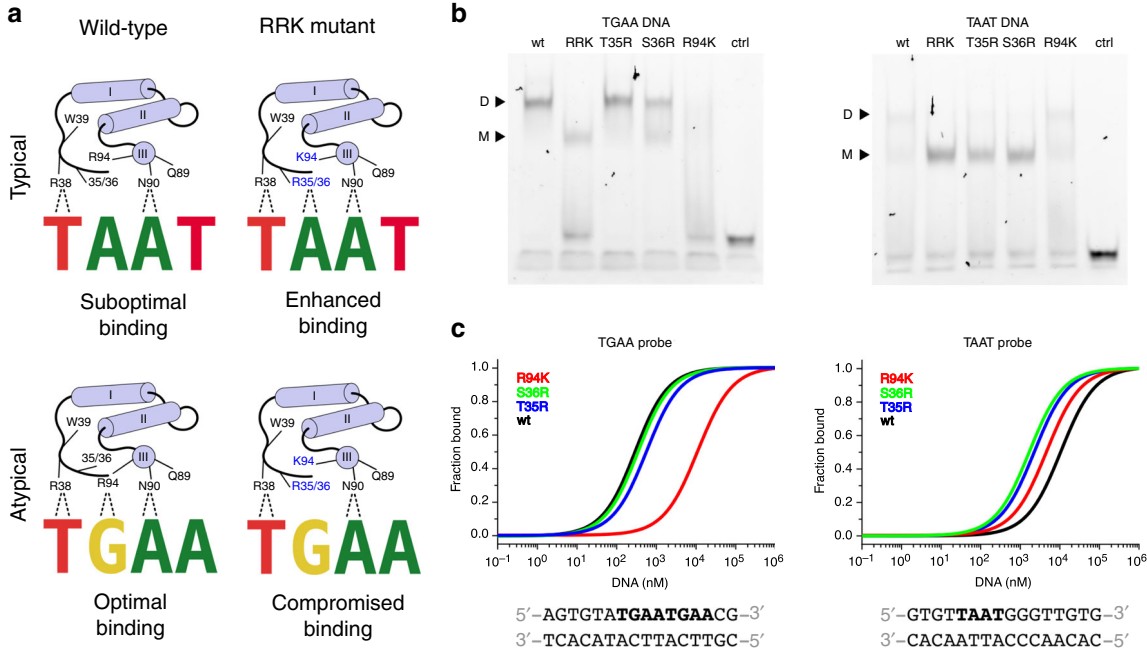

**Fig. 6 Altered DNA specificity of the WUS-HD. a** Schematic representation showing the sequence specificity of WUS-HD wild-type (left) and the RRK mutant (right) for typical TAAT (top panel) and atypical TGAA recognition motifs (bottom panel). Hydrogen bond interactions involved in base-recognition are highlighted (dashed lines) and relevant residues are indicated. Altered residues in the RRK mutant are shown in blue. **b** Electrophoretic mobility shift assays (EMSAs) of altered DNA specificity of WUS-HD for TGAA (left) and TAAT (right) probes. Monomer (M) and dimer (D) bound forms of WUS are indicated and the tested construct is given on top of the gel. **c** Quantification of WUS-HD DNA-binding affinity for TGAA (left) and TAAT (right) DNA by MST for the single point mutants T35R (blue), S36R (green), and R94K (red). As a reference the DNA-binding affinity of WUS-HD wt is shown in black.

unique orientations relative to each other for all motifs and the underlying protein–protein interactions are only partially overlapping. This finding is supported by other studies, which showed that the relative binding orientation of protein dimers can vary when bound to different DNA sequences[42]. The crystal structures of WUS in complex with DNA sequences from the *AG* (TAAT) and *CLV1* (TGAA and G-Box) promotor regions also show that WUS-HD binds both typical and atypical recognition motifs with similar residues, although the amino acid configuration strongly prefers the atypical TGAA motif over the typical TAAT motif. For example, R94 in the recognition helix can establish two hydrogen bonds with a guanine base, which leads to a strong preference for the atypical TGAA repeat sequence observed in EMSA and MST experiments. Further experiments demonstrated that mutagenesis of three key residues, namely T35 and S36 in the N-terminal arm and R94 in the C-terminal recognition helix, could shift WUS-HD sequence specificity from TGAA to TAAT recognition. These findings for the first time provide a mechanistically founded starting point to explore the contribution of the different binding motif interactions for in vivo function of WUS.

Members of WUS and TALE subfamily of HD-TFs in plants have been shown to form homo- and heterodimers[18,43]. Besides a downstream sequence (residues 134–208) important for dimerization[4,18,44], it was suggested that the HD also confers dimerization[44]. However, in this study MALS analysis of the isolated WUS-HD from *Arabidopsis thaliana* clearly demonstrated that this domain is a monomer in solution in contrast to the full-length protein that exhibited dimerization in yeast two-hybrid and FRET assays[4,18]. Consistently, previous analyses of the DNA-binding behavior of WUS-HD to TAAT motifs have shown that it binds DNA as monomer at lower concentrations and as dimer at higher levels, suggesting that dimerization is dependent on the presence of DNA and higher protein concentration[21].

These findings are also consistent with the two DNA-bound WUS-HD molecules we observed in the TGAA, G-Box, and TAAT crystal structures. While in the preferred arrangement on the TAAT motif we did not observe an interaction between the WUS-HD molecules, our structures and functional assays clearly identified the residues responsible for dimer formation for all other configurations. Interestingly, they do not match the suggestion made by Rodriguez et al.[44], who proposed that the interaction is dependent on G77. Since their experiments relied on yeast two-hybrid assays only, it is likely that a more general conformational change caused by changing G77 to E was responsible for the loss of homotypic interaction, rather than specific disruption of the dimer interface.

The important role of DNA for homodimerization of the isolated WUS-HD was further strengthened by our MST analysis which showed that both arrangement and orientation of specific recognition motifs are critical determinants for high affinity binding of WUS-HD (Fig. 7). For example, high affinity binding of WUS is dependent on appropriately arranged sequence motifs in a direct tandem repeat, as seen in the TGAA structure, with the dimerization interface being governed by hydrophobic interactions involving only a few amino acids. Changing this binding code in the DNA motif results in a drastic decrease of WUS-HD affinity.

In general, dimerization of TFs synergistically enhances the DNA-binding affinity relative to the interaction of each monomer alone[45]. Furthermore, dimerization increases the effective length of the recognized DNA sequence, which helps to discriminate between functional regulatory elements and randomly occurring isolated motifs that will not lead to productive gene regulation. Notably, in the case of the TGAA homodimer, which showed the highest binding affinity, the HDs associate non-symmetrically via the C-terminal end of the recognition helix (α3), thereby

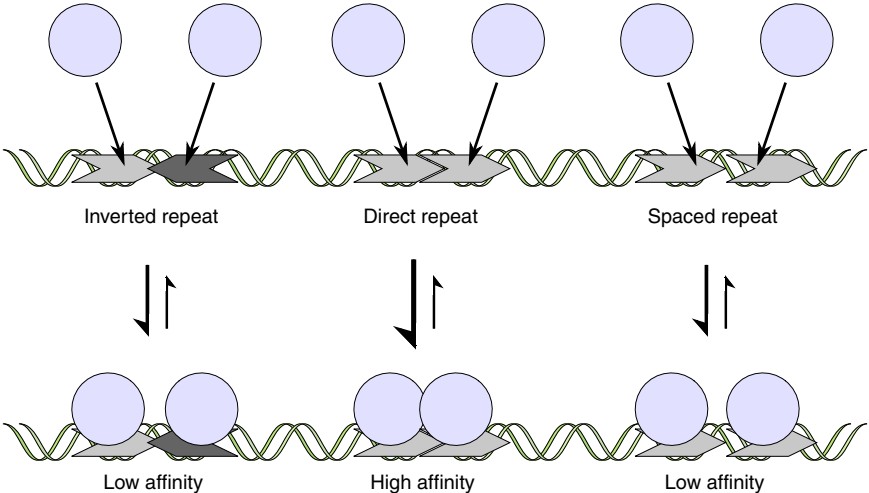

**Fig. 7 DNA-binding behavior of WUS-HD.** Schematic representation summarizing the complex DNA-binding behavior of the *At* WUS-HD to tandemly arranged recognition motifs. The WUS-HD (light blue) binds to specific DNA recognition sequences (gray), which can differ in their orientation and arrangement relative to each other. In the case of direct repeat sequences DNA-binding of the HD facilitates WUS-homodimerization via specific protein-protein interactions, leading to a cooperative gain in binding affinity. In contrast, inverted tandem repeats or direct repeats separated by a spacer do not allow protein–protein interactions to begin with and therefore are recognized with much lower affinity.

establishing additional contacts with the DNA. In contrast, the only other known example of a HD homodimer, the complex of the paired (PAX) HD bound to DNA, shows a symmetric head-to-head arrangement[46] (Supplementary Fig. 15a). Here, the N-terminal arm of each HD contacts the N-terminal end of helix α2 through complementarity of shape and charge with the underlying residues (Supplementary Fig. 15b). The symmetrical contacts between the two HDs lead to cooperative binding of a palindromic DNA sequence composed of two inverted TAAT motifs with high affinity. Further analysis of the DNA-binding activity of WUS-HD by MST and ITC demonstrated a cooperative gain in binding affinity with increasing number of TGAA recognition motifs and indicated a stabilization of the DNA interaction upon dimerization. Therefore, the specific protein–DNA and protein–protein interactions that contribute to the overall docking geometry all contribute to the final sequence recognition specificity of WUS. Importantly, these biochemical properties were mirrored by the binding behavior of native WUS to chromatin in plant cells, suggesting that our findings hold promise to decipher the regulatory potential of WUS in vivo.

A recent study analyzed the global effect of cytosine methylation on TF binding[47] and showed that many TFs, particularly the extended HD family, prefer CpG methylated sequences. Structural analysis highlighted that the specificity for methylcytosine depends on direct hydrophobic interactions with the 5-methyl group. Interestingly, analysis of the DNA contacts in the crystal structures of WUS bound to the TGAA and G-Box probes showed that the WUS-HD establishes hydrophobic contacts in a similar way to the methyl group of thymine. Furthermore, the bases at this position have a significant influence on the local DNA shape, with a narrow DNA MGW in the unbound state correlating with high affinity DNA-binding of WUS by the insertion of the conserved R38 into the minor groove. These findings are consistent with other examples of HD-DNA interactions showing that local optima in the DNA structure are preferentially recognized[9,36]. Besides DNA methylation, other epigenetic modifications, such as histone modifications, dynamically control gene expression, and thereby play an important role in maintaining cellular identity[48]. In particular, it is generally believed that the methylation of TF-binding sites is involved in cell proliferation and differentiation[47]. Emerging lines of evidence

also indicate that the aforementioned epigenetic mechanisms play vital roles in meristem maintenance and termination[49].

Another important level of control is post-translational modifications (PTMs) of TFs that can affect protein–protein interactions or modulate DNA-binding affinities. So far more than 200 different types of PTMs have been identified, including acetylation, glycosylation, and phosphorylation[50]. Numerous PTMs specifically modulate the interaction characteristics by modifying the electrostatic or structural properties of the protein[51]. Hence, PTM sites often correlate with protein interaction surfaces and frequently regulate key molecular processes. In line with these observations, the π-helix at the C-terminal end of helix α1 represents a potential PTM site for the regulation of WUS activity. The insertion of a tyrosine residue at position 54 is a WUS-specific feature within the WOX family that leads to a distortion at the end of helix α1. As π-helices are energetically disfavored they are usually associated with specific biological functions that are actively maintained during evolution[24,25]. One possibility could be that Y54 represents a regulatory switch, where phosphorylation of the hydroxyl-group would alter WUS DNA-binding specificity. A structural model suggests that R94 could be involved in the coordination of the phosphate group of pY54 (Supplementary Fig. 16). In addition, H91 and the DNA phosphate-backbone could stabilize this interaction, thereby altering the sequence specificity and the overall DNA-binding affinity.

In summary, the structural and biochemical analyses of the complex DNA-binding mechanism of WUS-HD presented here provide insight into the molecular basis of the underlying sequence recognition specificity of WUS and will help to dissect the complex regulatory network controlling stem cell fate in plant meristems. The crystal structures of WUS bound to different DNA probes determined in this study highlight important mechanistic details of specific sequence recognition. In particular, homodimerization of WUS upon DNA-binding represents one of the key determinants to achieve high affinity binding of specific regulatory elements (Fig. 7), despite the broad sequence specificity generally observed for HDs. Furthermore, this cooperative DNA-binding mechanism governed by the association of two HDs likely represents a general mechanism of the plant-specific WOX protein family to recognize direct repeat sequences of the

atypical TGAA motif with high specificity. Nevertheless, further experiments are required to uncover the physiological role of WUS homodimerization for DNA-binding in vivo and its functional significance for gene regulation in the shoot meristem.

## Methods

**Cloning**. The WUS-HD (34–103) was cloned into pETMBP[52] and pETYFP via NcoI/XhoI restriction sites. The pETYFP vector was generated by cloning the gene encoding YFP into pETM11 with a C-terminal StrepII-tag. The resulting YFP constructs are C-terminal fusions to a His6-tagged YFP protein, with an additional C-terminal StrepII-tag. The resulting MBP constructs are C-terminal fusions to a His6-tagged MBP protein with a cleavable tobacco etch virus (TEV) site. Point mutations in pETMBP_WUS-HD and pETYFP_WUS-HD were generated using the QuikChange system (Stratagene). To create the 6xHis-tagged MBP fusion of full-length WUS the coding sequence was amplified with Phusion DNA Polymerase (Thermo Fisher Scientific), digested with NcoI/XhoI and ligated into pMG210[53] previously digested with NcoI/SalI. All primers used for cloning are listed in Supplementary Table 1.

**Protein production and purification**. Protein expression was carried out in Rosetta2 (DE3) cells either by IPTG induction (1 mM final concentration) using LB medium or auto-induction based on the protocol by Studier[54]. Cells were harvested in cold lysis buffer (20 mM TRIS/HCl (pH 8.0), 150 mM NaCl, 10 mM imidazole, 0.02% 1-thioglycerol, 1 mg/ml lysozyme, 1 mg/ml DNase, EDTA-free protease inhibitor cocktail (Roche)). The cell suspension was homogenized by four passes through a Microfluidizer (M1-10L, Microfluidics). The lysate was cleared by centrifugation at 50,000×g and filtered through a 0.45 μm filter before application to a HisTrap FF column (GE Healthcare). The column was washed with IMAC buffer (20 mM TRIS/HCl (pH 8.0), 150 mM NaCl, 10 mM imidazole, 0.02% 1-thioglycerol) and the same buffer containing 1 M NaCl. The protein was eluted with IMAC buffer containing 330 mM imidazole and loaded onto a HiTrap SP HP column (GE Healthcare). The column was washed with ion exchange buffer buffer (20 mM TRIS/HCl (pH 8.0), 150 mM NaCl, 1 mM DTT) and eluted with the same buffer containing 500 mM NaCl. The final eluate was subjected to size-exclusion chromatography using a 16/60 Superdex 75 column (GE Healthcare) equilibrated in gel filtration buffer (20 mM TRIS/HCl (pH 8.0), 150 mM NaCl, 2 mM DTT). Finally, the protein was exchanged into storage buffer (20 mM TRIS/HCl (pH 8.0), 75 mM NaCl, 10 mM MgCl2) using a PD-10 desalting column (GE Healthcare) and aliquots were snap-frozen in liquid nitrogen and stored at −80 °C. Purification of the MBP fusion constructs included the following adaptations: after IMAC purification an additional TEV cleavage (1:100) was performed overnight at 4 °C in gel filtration buffer for MBP WUS-HD and for MBP WUS-FL the size-exclusion step was omitted and the protein was concentrated directly after the desalting step.

**Oligonucleotide assembly**. Duplex DNAs were annealed using complementary single-strand HPLC purified DNAs (Eurofins). Lyophilized oligonucleotides were dissolved in crystallization buffer (20 mM TRIS/HCl (pH 8.0), 75 mM NaCl, 10 mM MgCl2, 2 mM DTT) or MST buffer (20 mM TRIS/HCl (pH 8.0), 75 mM NaCl, 10 mM MgCl2, 0.05 % Tween-20) and mixed 1:1 with the complementary strand at a final concentration of 50–500 μM. The reaction mix was heated to 95 °C for 5 min and cooled over 75 cycles, each cycle lasting 1 min and decreasing the temperature by 1 K.

**Protein crystallization and data collection**. The WUS-HD alone (6–12 mg/ml) was crystallized in an in-house automated crystallization platform. The best crystals were obtained after 1–3 days at 18 °C in sitting drops containing 0.05 M MES (pH 6.0), 0.01 M Mg(OAc)2 and 2.5 M AmSO4. For each complex, the WUS-HD was first mixed with a solution of annealed, blunt-ended DNA duplex (16-bp) at a molar ratio of 1:0.6 (protein:DNA) and after 30 min on ice subjected to crystallization trials. The crystallization conditions for all complexes were optimized using an in house developed crystal screening kit for PEG conditions. Complexes were crystallized in sitting drops by vapor diffusion technique from solution containing different concentrations of various PEGs. Crystals with G-Box DNA were grown in 0.2 M LiOAc and 20% PEG3350. The TGAA DNA crystals were grown in 0.1 M NaOAc (pH 4.6) and 8% PEG4000. Crystals with TAAT DNA were grown in 0.1 M CHES (pH 9.5) and 20% PEG8000. All crystals were cryoprotected in mother liquor containing 20% (v/v) glycerol and flash-cooled in liquid nitrogen. Data sets were collected at the ESRF from individual crystals on beam-lines ID30-A3 (WUS-HD and WUS-HD + G-Box DNA) and ID23-1 (WUS-HD + TAAT DNA and WUS-HD + TGAA DNA) at 100 K. Oligonucleotide sequences are listed in Supplementary Table 2.

**Structure determination and refinement**. The WUS-HD alone crystallized in the tetragonal space group P 41212 with one molecule in the asymmetric unit. The cell parameters are a = 43.1 Å, b = 43.1 Å, c = 82.0 Å and α, β, γ = 90 °C. The WUS-HD structure was solved by molecular replacement as implemented in PHASER[55] using the HD of Engrailed (PDB code 3HDD[23]) as a search model. After several rounds of manual building in Coot v0.9[56] the structure was refined with Phenix

v1.16[57]. The model quality was analyzed with MOLPROBITY[58] giving Ramachandran statistics for the final model of 98.4% of residues in favored regions, 1.6% in allowed regions and 0% outliers.

All WUS–DNA complex structures were solved by molecular replacement as implemented in PHASER[55] using the WUS-HD as a search model. Structure refinement was performed with Phenix v1.16[57] and iterative model building in Coot v0.9[56]. WUS in complex with G-Box DNA crystallized in the monoclinic space group C 121 with seven molecules in the asymmetric unit. The cell parameters are a = 123.4 Å, b = 83.1 Å, c = 85.7 Å and α, β = 90 °C, γ = 113 °C. The model quality was analyzed with MOLPROBITY[58] giving Ramachandran statistics for the final model of 97.9% of residues in favored regions, 1.7% in allowed regions and 0.3% outliers. WUS in complex with TGAA DNA crystallized in the monoclinic space group P 1211 with six molecules in the asymmetric unit. The cell parameters are a = 55.7 Å, b = 54.3 Å, c = 75.6 Å and α, β = 90 °C, γ = 107.5 °C. The model quality was analyzed with MOLPROBITY[58] giving Ramachandran statistics for the final model of 100% of residues in favored regions, 0% in allowed regions, and 0% outliers. WUS in complex with TAAT DNA crystallized in the monoclinic space group P 1211 with seven molecules in the asymmetric unit. The cell parameters are a = 82.7 Å, b = 45.9 Å, c = 87.5 Å and α, β = 90 °C, γ = 102.9 °C. The model quality was analyzed with MOLPROBITY[58] giving Ramachandran statistics for the final model of 99.3% of residues in favored regions, 0.7% in allowed regions, and 0% outliers. The data collection and refinement statistics for all structures are summarized in Table 1.

**Multi-angle light scattering (MALS)**. Experiments were performed using in-line size exclusion chromatography coupled to MALS and differential refractive index (dRI) measurements. The WUS-HD was injected on a Superdex 75 10/300 GL gel filtration column (GE Healthcare) equilibrated with gel filtration buffer (20 mM TRIS/HCl (pH 8.0), 75 mM NaCl, 10 mM MgCl2, 2 mM DTT). The chromatography system was coupled to a DAWN HELEOS II detector and Optilab T-rEX dRI detector (both Wyatt Technology). Data analyses were performed with the ASTRA V software using a dn/dc value of 0.185 mg/ml for molar mass calculation.

**Isothermal titration calorimetry (ITC)**. All samples were dialyzed overnight at 4 °C against ITC buffer (20 mM TRIS/HCl (pH 8.0), 75 mM NaCl, 10 mM MgCl2). ITC experiments of WUS-HD and a 16-bp DNA 2xTGAA oligonucleotide were performed using a PEAQ-ITC microcalorimeter (Malvern) at 20 °C. Titrations consisted of 13–19 injections of 2–3 μl aliquots of the titrant into the cell solution and 150 s intervals between injections. Typical concentrations used were 400–500 μM WUS-HD in the syringe and 20–40 μM DNA in the cell. Data evaluation were performed with the PEAQ-ITC analysis software.

**Microscale thermophoresis (MST)**. Binding affinities of WUS-HD to different DNA sequences was measured using MST. For MST measurements WUS-HD was prepared as a C-terminal fusion to YFP, taking advantage of the intrinsic fluorescence. The YFP–WUS fusion was diluted into MST buffer (20 mM TRIS/HCl (pH 8.0), 75 mM NaCl, 10 mM MgCl2, 0.05% Tween-20) prior to use. A dilution series of 16-bp target DNA in MST buffer was prepared and mixed 1:1 with the protein. Typical DNA concentrations used were 0.001–100 μM and a fixed concentration of 50 nM for YFP–WUS. Protein–DNA binding reactions were loaded into Monolith NT.115 Premium Capillaries (NanoTemper Technologies) and the thermophoretic response was monitored with 30% infrared (IR) laser intensity and 80% MST power using a Monolith NT.115 instrument (NanoTemper Technologies) at 20 °C. The normalized change in fluorescence was plotted as a function of DNA concentration and analyzed using the MO.Affinity Analysis software (NanoTemper Technologies). Oligonucleotide sequences are listed in Supplementary Table 3.

**Electrophoretic mobility shift assay (EMSA)**. EMSAs were carried out as described in Brackmann et al.[53]. Each reaction contained 115 pmol WUS-HD and 200 fmol CY5-labeled probe in 10 mM TRIS/HCl (pH 8.0), 75 mM NaCl, 50 μg/ml Poly(dI-dC), 1 mM EDTA, 1 mM DTT, 5% glycerol. Oligonucleotide sequences are listed in Supplementary Table 4.

**Bioinformatics**. Alignments were generated with Clustal Omega[59] and visualized with ESPript v3.0[60]. Surface representations of conserved residues were generated using the ConSurf server[61]. All structural figures were prepared with PyMOL v2.3.2[62]. Electrostatic surface potentials were calculated with APBS[63] integrated in PyMOL. The atomic displacement parameters (residue average) were calculated with BAVERAGE from the CCP4 package v7.0[64]. Superimpositions were calculated with GESAMT[65] from the CCP4-package. The interaction parameters and surface area were calculated with PISA[66] from the CCP4 package.

As a in vivo affinity measure for native WUS protein to chromatin containing the three binding sites studied here, we analyzed ChIP-seq data from *Arabidopsis* seedlings with ectopically induced WUS expression[28]. As a proxy for affinity we used the number of reads aligned to regions containing WUS-binding sequences represented as a specific k-mer. For this purpose, we created BED files using the *A. thaliana* (TAIR10) genome with the binding motifs under investigation. We extracted genomic coordinates (chromosome, start, end) from the TAIR10

annotation for each occurrence of a specific binding motif and defined a window of ±25 bp around the sequence coordinates using GenomicRanges v1.32.6 R-package[67]. The resulting genomic regions were transformed into GTF files and used for counting aligned reads from WUS ChIP-seq data (GEO accession GSE122611) with featureCount v1.6.3[68]. The obtained counts per binding motif containing window were visualized by empirical cumulative probability density using ecdf function in R v3.6.2 and smoothed using plogspline function from R-package logspline v2.1.15[69]. Further details and interactive datasets are available in Supplementary File 1.

**Reporting summary**. Further information on research design is available in the Nature Research Reporting Summary linked to this article.

## Data availability

Crystal structure coordinates and structure factors have been deposited to the Protein Data Bank (PDB; www.rcsb.org) under the accession codes 6RY3 (WUS-HD), 6RYI (WUS-HD + G-Box DNA), 6RYL (WUS-HD + TAAT DNA), and 6RYD (WUS-HD + TGAA DNA). The data underlying Figs. 2a and 6b as well as Supplementary Figs. 3, 7, 9, 10, and 14 are provided as a Source Data file. Any other data supporting the findings of this study are available from the corresponding authors upon reasonable request.

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

## Acknowledgements
We thank J. Kopp and C. Siegmann from the BZH/Cluster of Excellence: CellNetworks crystallization platform for protein crystallization; N. Dobrev for expression plasmids; Y.L. Ahmed for crystallographic support and advice during the early stages of the project; S. Zhang for support during purification of full-length WUS and EMSAs; M. McDowell for fruitful discussions and comments on the paper. We acknowledge the ESRF for support and access to the beamlines. I.S. and J.U.L. are investigators of the CellNetworks Cluster of Excellence. This work was supported by the DFG through the Leibniz program (SI 586/6-1 to I.S.) and the SFB873 to J.U.L.

## Author contributions
J.S., J.P.H., I.S., and J.U.L. conceived the study. J.S., J.P.H., M.G., A.G., and G.S. generated constructs and purified proteins. J.S., J.P.H., and A.G. prepared samples for crystallography and collected data. J.S., A.G., and K.W. determined crystal structures, J.S., K.W., and I.S. analyzed the structural data. J.S., J.P.H., and M.G. performed the binding experiments. O.E. performed the bioinformatic analysis; J.S., I.S., and J.U.L. wrote the paper. All authors discussed and commented on the final paper.

## Competing interests
The authors declare no competing interests.
