## [Peer Review File · Nature Communications]

Reviewers' comments:

Reviewer #1 (Remarks to the Author):

WUSCHEL (WUS) is a plant transcription factor playing a central role in the stem cell niche in *Arabidopsis* as well as in many other species. It belongs to the homeodomain (HD) family of transcription factors (TF) and was shown to induce and repress gene expression. Conflicting data have been published regarding the DNA sequences (TAAT core, a G-Box like and a TGAA repeat element) that are recognised by WUS depending on the technique used (ChIP-seq, Selex or DAP-seq).

In this manuscript, the authors use a combination of crystallographic and biochemical/biophysical characterisation to describe the complex between the isolated WUS HD and several DNA sequences. They solve the structure of the apo version WUS HD (that shows small differences with classical HD) and of WUS HD in complex with the 3 types of binding sites. The main message is that WUS binds DNA in different ways, sometimes involving head-to-tail dimerisation that was never observed before and should increase the specificity. This represents an important advance for this key transcription factor.

The significance of these structural findings would be very high if it was well established that WUS was regulating the expression of its target genes as homodimer through one of these 3 cis-elements, and if the effect of point mutations (affecting DNA binding or specific modes of dimerisation) could be tested on the regulation of these different targets.

As such,

- * The triple (probably not just double according to O'Malley et al.) TG[A/T][A/T] is the highest affinity binding site, it was found in vitro using genomic DNA but not validated in planta.
- * The G-box is the best characterised element as it is found in the *CLAVATA1* gene and it was shown to mediate repression by WUS in a transient assay.
- * The TAAT was found within a binding site for the *LEAFY* homeotic protein in the *AGAMOUS* floral gene (Lohmann et al 2001). However, I could not find solid evidence that WUS truly binds to this element. From subsequent publications on *LEAFY*, it appears that the TAAT mutation is expected to affect the binding of *LEAFY* (Winter et al. 2011, Moyroud et al. 2011) and therefore the response to *LEAFY*+WUS but not necessarily the WUS binding itself. Thus, there is little evidence that the TAAT is the WUS response element in plants.

Another point worth noting is that WUS was shown to be able to dimerise (Busch et al. 2010) and the different experiments (DAP/Selex/ChIP) have been done with such a protein but the structural and biochemical characterisations were done on the isolated HD. Because the manuscript mainly deals with various mode of dimerisation, it is quite important to test some of these properties in the context of the whole protein.

More technical concerns

- * for several of the WUS binding sites, the importance of individual bases has not been tested in solution. When removing a WUS HD monomer from the structure because it is a likely crystal packing artefact, it is important to validate that the remaining protein-DNA contacts are those present in solution. This is most problematic case is for the structure on a G-box motif. The monomer that was removed is in the same position as the one kept on the TAAT complex structure. The rationale for keeping one and removing the other is not very clear and affinity measurements with mutant DNA would help to distinguish.
- * the absence of one HD monomer in MALS might indeed be due to dilution effect during SEC but it could also suggest that the crystallographic complex shows additional monomers due to high concentration in the crystal solution. Validating the crystallographic data in solution is again important.

In conclusion, I think this manuscript includes a very interesting and detailed biochemical and structural analysis. Still, it could be strengthened with more biochemical data (including number of the proteins per complex and mutation of individual bases to validate crystal based results). But my main concern is that the biological significance of the binding sites and protein/DNA complexes analysed remains quite elusive because the data on the WUS binding sites are too fragmented to justify the detailed structural characterisation presented here. The absence of solid target gene for each element also precludes any test in planta to validate the biochemical findings using reporter assays.

Reviewer #2 (Remarks to the Author):

The paper reported the biochemical and structural studies on the recognition of different cis-elements by plant HD protein WUS. Generally, the experiments were performed with high standard and the paper was well written. However, the solely structure analysis lacks functional supports. Considering the HD is an extensively studied DNA binding domain, the novelty of the paper is not high. Some functional studies are highly recommended to connect the structural observations and the in vivo function.

Point-by-point response

Reviewer #1:

>> WUSCHEL (WUS) is a plant transcription factor playing a central role in the stem cell niche in Arabidopsis as well as in many other species. It belongs to the homeodomain (HD) family of transcription factors (TF) and was shown to induce and repress gene expression. Conflicting data have been published regarding the DNA sequences (TAAT core, a G-Box like and a TGAA repeat element) that are recognised by WUS depending on the technique used (ChIP-seq, Selex or DAP-seq).

In this manuscript, the authors use a combination of crystallographic and biochemical/biophysical characterisation to describe the complex between the isolated WUS HD and several DNA sequences. They solve the structure of the apo version WUS HD (that shows small differences with classical HD) and of WUS HD in complex with the 3 types of binding sites. The main message is that WUS binds DNA in different ways, sometimes involving head-to-tail dimerisation that was never observed before and should increase the specificity. This represents an important advance for this key transcription factor.

-> We thank the reviewer for pointing out the novelty and importance of our work.

>> The significance of these structural findings would be very high if it was well established that WUS was regulating the expression of its target genes as homodimer through one of these 3 cis-elements, and if the effect of point mutations (affecting DNA binding or specific modes of dimerisation) could be tested on the regulation of these different targets.

-> Apart from the TGAA repeat element, which only recently has been identified, there is substantial evidence for gene regulatory activity of WUS through the DNA motifs we studied (e.g. Lohmann et al. Cell 2001; Leibfried et al. Nature 2005, Busch et al. Developmental Cell 2010; Perales et al. PNAS 2016; Rodriguez et al. PNAS 2016). Some of these publications specifically address the role of homo-dimerization and its role for target gene regulation, albeit without biochemical or structural data. We have now included this information in the introduction.

>> As such,

* The triple (probably not just double according to O'Malley et al.) TG[A/T][A/T] is the highest affinity binding site, it was found in vitro using genomic DNA but not validated in planta.

* The G-box is the best characterised element as it is found in the CLAVATA1 gene and it was shown to mediate repression by WUS in a transient assay.

* The TAAT was found within a binding site for the LEAFY homeotic protein in the AGAMOUS floral gene (Lohmann et al 2001). However, I could not find solid evidence that WUS truly binds to this element. From subsequent publications on LEAFY, it appears that the TAAT mutation is expected to affect the binding of LEAFY (Winter et al. 2011, Moyroud et al. 2011) and therefore the response to LEAFY+WUS but not necessarily the WUS binding itself. Thus, there is little evidence that the TAAT is the WUS response element in plants.

-> WUS was described to bind a number of diverse TAAT elements in the regulatory regions not only of AG, but also of ARR7 (Leibfried et al. Nature 2005), CLV3 (Yadav et al. 2011, Perales et al. 2016), KAN1 (Yadav et al. 2013) and in the promoter of GRP23 (Zhang et al. 2013). In all cases direct binding to this element and/or relevance for proper reporter gene activity was shown to be mediated by WUS. We thank the reviewer for pointing out that we need to state these facts more clearly in our manuscript, which we now have.

>> Another point worth noting is that WUS was shown to be able to dimerise (Busch et al. 2010) and the different experiments (DAP/Selex/ChIP) have been done with such a protein but the structural and biochemical characterisations were done on the isolated HD. Because the manuscript mainly deals with various mode of dimerisation, it is quite important to test some of these properties in the context of the whole protein.

-> We agree that comparing the binding behaviour of the WUS-HD to the one of full-length WUS is important and therefore have now carried out a number of additional experiments to address this: On the one hand, we have performed EMSA experiments with full-length WUS fused to MBP that confirm our findings from the WUS HD alone (Supl Fig. 3) by showing a clear difference in binding to TAAT, G-Box and TGAA DNA probes. Even at the quantitative level, the affinities from these new experiments correlate well with our MST data using the WUS-HD. On the other hand, we have analysed WUS ChIP-seq data to quantify WUS chromatin binding in vivo. Again, the relative affinities derived from our analysis mirror the data obtained with the WUS-HD in vitro. Taken together, we found a very strong correlation between experiments using WUS-HD and full length WUS, both in vitro and in vivo.

>> More technical concerns:

* for several of the WUS binding sites, the importance of individual bases has not been tested in solution. When removing a WUS HD monomer from the structure because it is a likely crystal packing artefact, it is important to validate that the remaining protein-DNA contacts are those present in solution. This is most problematic case is for the structure on a G-box motif. The monomer that was removed is in the same position as the one kept on the TAAT complex structure. The rationale for keeping one and removing the other is not very clear and affinity measurements with mutant DNA would help to distinguish.

-> We agree with this reviewer and have carried out a number of additional experiments and revisions to the text to address this point:

We have performed additional EMSA experiments with different versions of the TAAT probe (Supl Fig. 7) to test which of the two additional WUS molecules is most relevant in solution. By introducing mutations into the probe that only affect binding of a specific WUS HD, we now find that indeed both binding configurations are relevant in solution. We now demonstrate that the protein-DNA contact on the opposite side of the TAAT motif is preferred in solution, although only present in one of the crystal complexes. In contrast, the second additional DNA contact site present in both crystal complexes is less relevant for WUS-binding in solution. Thus, dimerisation between two WUS HDs is less important for WUS-binding to TAAT in solution than it is to TGAA although the observed dimer interface is much larger.

With regards to the interaction of WUS with the G-Box we have now clarified that systematic EMSA experiments on mutated G-Box sequences have already been published and show quantitatively which protein-DNA contact sites from crystallography contribute to probe binding in solution (Busch et al. 2010). Furthermore, initial crystal structures of the WUS G-Box complex only included two HDs per DNA molecule (similar to Supl Fig. 5). However, these crystals only diffracted poorly (resolution >3 Å). We therefore concluded that the additional third HD in the G-Box structure aided crystal contact formation and thus likely reflects a crystal packing artefact. This information has been added to the manuscript.

* the absence of one HD monomer in MALS might indeed be due to dilution effect during SEC but it could also suggest that the crystallographic complex shows additional monomers due to high concentration in the crystal solution. Validating the crystallographic data in solution is again important.

-> In agreement with reviewer #1, we frequently observed higher molecular weight shifts during EMSA experiments when we added high protein concentrations of WUS-HD, highlighting the broad sequence specificity of HDs, which can lead to DNA-binding of additional monomers. However, determining the absolute oligomeric state of the complex is difficult as we have shown in solution by ITC, because the dissociation constants for individual HD-binding events are very similar. In addition, we have tested DNA probes with varying number of recognition sites in our MST experiments and observed a good correlation between the increase in binding affinity and the number of DNA-binding sites (Supl Fig. 12). We have clarified this in the text.

>> In conclusion, I think this manuscript includes a very interesting and detailed biochemical and structural analysis. Still, it could be strengthened with more biochemical data (including number the proteins per complex and mutation of individual bases to validate crystal-based results). But my main concern is that the biological significance of the binding sites and protein/DNA complexes analysed remains quite elusive because the data on the WUS binding sites are too fragmented to justify the detailed structural characterisation presented here. The absence of solid target gene for each element also precludes any test in planta to validate the biochemical findings using reporter assays.

-> As pointed out above, WUS target genes are already published with AG, *CLV3* and *GRP23* for TAAT (Lohmann et al. 2001, Yadav et al. 2011, Perales et al. 2016, Zhang et al. 2013), *CLV1*, *TPL*, *TPR1* for G-Box (Busch et al. 2010, Ma et al. 2019) and *CLV1* and *TPL* for 2xTGAA elements (Ma et al. 2019, see image below). Because reporter assays in transgenic lines are inherently noisy and only allow for qualitative readouts, we have chosen to quantitatively analyse WUS ChIP-seq data to test the relevance of our biochemical data in vivo.

Using the number of WUS ChIP-chip sequencing reads that co-occur with the TAAT, G-Box and TGAA motif as a proxy for affinity, we have now analysed all biochemically tested variants for their WUS occupancy in vivo. These data clearly show a higher binding probability of WUS for the TGAA repeat motif over G-Box and TAAT. Importantly, even different arrangements of TGAA target sites exhibited differential binding, consistent with our *in vitro* MST and EMSA data. These genome-wide and unbiased analyses taking many thousand binding sites into account demonstrate the *in vivo* relevance of the binding sites studied biochemically.

Reviewer #2 :

>> The paper reported the biochemical and structural studies on the recognition of different

cis-elements by plant HD protein WUS. Generally, the experiments were performed with high standard and the paper was well written. However, the solely structure analysis lacks functional supports. Considering the HD is an extensively studied DNA binding domain, the novelty of the paper is not high. Some functional studies are highly recommended to connect the structural observations and the in vivo function.

-> We agree with reviewer #2 that HD proteins in general are well-studied model systems for DNA-binding. However, WUS has been reported to bind to diverse DNA motifs and to act as transcriptional activator and repressor and the mechanisms underlying this remarkable behavior have remained unclear. Taken together, our structural and biochemical results highlight a novel mode of binding important for high affinity interaction. Consistent with a comment from reviewer #1 *“The main message is that WUS binds DNA in different ways, sometimes involving head-to-tail dimerisation that was never observed before and should increase the specificity. This represents an important advance for this key transcription factor.”* these findings provide a mechanistic basis for dissecting WUS-dependent regulatory networks.

We have now included computational analyses of ChIP-seq data that provide robust functional support for our findings made in vitro (see also response to reviewer #1).

We are also aware that further experiments are required to uncover the physiological role of WUS homodimerization for DNA-binding *in vivo* and its functional significance for gene regulation in the shoot meristem, however, these experiments would take years and thus go well beyond the scope of this manuscript.

REVIEWERS' COMMENTS:

Reviewer #1 (Remarks to the Author):

I have now carefully read the revised version of the Sloan et al manuscript. Even if the main criticism raised by both reviewers in the first round has not really been addressed (lack of functional data conferring a biological significance to the detailed in vitro analysis), I acknowledge the efforts made to try improving the manuscript.

Better referencing the work from the Reddy lab is indeed a good idea as they indeed provide the most solid in vivo and in vitro evidence for the importance of TAAT WUS binding sites.

The various experiments that have been added do answer some of my concerns and bring some new and valuable information.

* The in vitro with the full length protein nicely back up work with the isolated DBD.

* Understanding where the different monomers are positioned on the DNA is an essential aspect of this work and I think it's good that additional experiments have been performed to address this point. Regarding the TAAT probe, the T4C mutation indeed reduces the binding consolidating the presence of the light blue monomer from figure 2 but the T12C and T12C T15C have no effect (very mild difference, impossible to interpret because the probes are different and free DNA are identical) and I think one can safely conclude that the teal monomer is a likely crystallographic artefact. I do not really understand the author's conclusion

* Using ChIP-seq data to derive affinity of the different types of binding sites is indeed a good idea.

However, I did not manage to understand what has been done. The procedure should be better described. The rebuttal mentions ChIP-Chip 'reads' whereas the rest of the manuscript refers to ChIP-seq. The methods section would deserve some proofreading! "For this purpose, we created BED files with using the of Arabidopsis thaliana (TAIR10) genome and k-mers of interest. We the identified regions with known WUS binding sites ». The k-mers used to discriminate the different binding sites have to be provided as a list to together with additional information explaining on which basis they have been sorted to discriminate the different classes (TAAT, TGAA, G-box).

Finally the method used to infer affinity based on the reads covering these k-mers remains cryptic to me even after reading the methods multiple times. Since the method is used several times and is the main response to both reviewers's comments, I believe it should be explicit and convincing! Overall, this work remains an interesting detailed biochemical and structural characterisation of the protein/DNA complex formed between the DNA binding domain of an important plant development regulator and several of its DNA targets. The links between these characterisations and the functions of this regulator remain fairly weak but I hope this work can serve as a basis to better understand how WUS performs its different roles in the plant in the future.

Reviewer #2 (Remarks to the Author):

The revised version responded most of the concerns from the reviewers, especially that the analysis of ChIP data can well reflect the biological significance of the biochemical data. I have no more concerns.

Point-by-point response to the reviewers:

Reviewer #1:

I have now carefully read the revised version of the Sloan et al manuscript. Even if the main criticism raised by both reviewers in the first round has not really been addressed (lack of functional data conferring a biological significance to the detailed in vitro analysis), I acknowledge the efforts made to try improving the manuscript.

Better referencing the work from the Reddy lab is indeed a good idea as they indeed provide the most solid in vivo and in vitro evidence for the importance of TAAT WUS binding sites. The various experiments that have been added do answer some of my concerns and bring some new and valuable information.

- **Response:** We thank the reviewer for the positive feedback on our revisions.

* The in vitro with the full length protein nicely back up work with the isolated DBD.

- **Response:** We thank the reviewer for the positive feedback on our revisions.

* Understanding where the different monomers are positioned on the DNA is an essential aspect of this work and I think it's good that additional experiments have been performed to address this point. Regarding the TAAT probe, the T4C mutation indeed reduces the binding consolidating the presence of the light blue monomer from figure 2 but the T12C and T12C T15C have no effect (very mild difference, impossible to interpret because the probes are different and free DNA are identical) and I think one can safely conclude that the teal monomer is a likely crystallographic artefact. I do not really understand the author's conclusion.

- **Response:** We have again very carefully analyzed our structural and EMSA data and agree with the reviewer that we cannot rule out that the teal WUS monomer shown on the TAAT probe in Figure 2 may represent a crystallization artefact. Since we do see some change in binding behavior in our EMSAs using the mutated probes, we have now modified the text and discussion to clarify that this configuration either plays only a minor role or is an artefact of crystal packing.

* Using ChIP-seq data to derive affinity of the different types of binding sites is indeed a good idea. However, I did not manage to understand what has been done. The procedure should be better described. The rebuttal mentions ChIP-Chip 'reads' whereas the rest of the manuscript refers to ChIP-seq. The methods section would deserve some proofreading ! "For this purpose, we created BED files with using the of Arabidopsis thaliana (TAIR10) genome and k-mers of interest. We the identified regions with known WUS binding sites ». The k-mers used to discriminate the different binding sites have to be provided as a list to together with additional information explaining on which basis they have been sorted to discriminate the different classes (TAAT, TGAA, G-box). Finally the method used to infer affinity based on the reads covering these k-mers remains cryptic to me even after reading the methods multiple times. Since the method is used several times and is the main response to both reviewers's comments, I believe it should be explicit and convincing!

Overall, this work remains an interesting detailed biochemical and structural characterisation of the protein/DNA complex formed between the DNA binding domain of an important plant development regulator and several of its DNA targets. The links between these characterisations and the functions of this regulator remain fairly weak but I hope this work

can serve as a basis to better understand how WUS performs its different roles in the plant in the future.

- **Response:** We agree with the reviewer that our description of the ChIP-seq analysis was too superficial. We have now included an explanatory paragraph in the main text and have modified the method section. In addition, we have now included a supplementary HTML file detailing the analysis including figures, code and interactive tables to make our data much more accessible.

Reviewer #2:

The revised version responded most of the concerns from the reviewers, especially that the analysis of ChIP data can well reflect the biological significance of the biochemical data. I have no more concerns.

- **Response:** We thank the reviewer for the positive feedback on our revisions.